# Why Healthy Pine Seedlings Die after They Leave the Nursery

David B. South [1,*], Tom E. Starkey [2] and Al Lyons [3]

1 College of Forestry, Wildlife Sciences and the Environment, Auburn University, Duncan Drive, Auburn, AL 36849, USA

2 Forest Nursery & Regeneration Consultant, 220 Northridge Dr, Anderson, SC 29621, USA; tomstarkey3@gmail.com

3 Manulife Investment Management, 3891 Klein Road, Harpersville, AL 35078, USA; alyons@manulife.com

* Correspondence: southdb@auburn.edu

**Abstract:** Artificial regeneration is successful when high-performing seedlings are transported with care to the planting site, stored for a short period in an environment without desiccation or fungal growth, and planted in a deep hole, so roots are in contact with moist soil. One of the requirements for success is the ability to avoid common planting mistakes. Due, in part, to the use of container stock plus an increase in rainfall, the average first-year survival of pine seedlings (89%) in the southern United States is about 15% greater now than 45 years ago. However, when survival is less than 50% six months after planting, some landowners seek reimbursement for their loss. Some assume poor seedling quality was the cause without realizing that anaerobic soils or sudden freeze events, shallow planting holes, pruning roots, a lack of rain or underground insects can kill pines. With a focus on pines planted in the southern United States, we list non-nursery factors that have killed seedlings in North America, Africa and Europe.

**Keywords:** planting depth; drought; freezing injury; herbivory; mortality; survival; insects

## 1. Introduction

When most factors are optimum, the 11-month survival of bareroot pine seedlings can exceed 90% [1,2], but in some years, survival is less than 75%. It seems the initial survival of pine seedlings in 13 southern United States (SUS) has gradually increased over time (Figure 1). The average survival of loblolly pine (*Pinus taeda* L.) for 58 sites during the 1980s was 74% [2,3], and now it averages about 89%. This increase is likely due to improvements in the seedling quality of bareroot stock combined with planting more container stock. In the SUS, container stock now represents one-fourth of the pine seedling production.

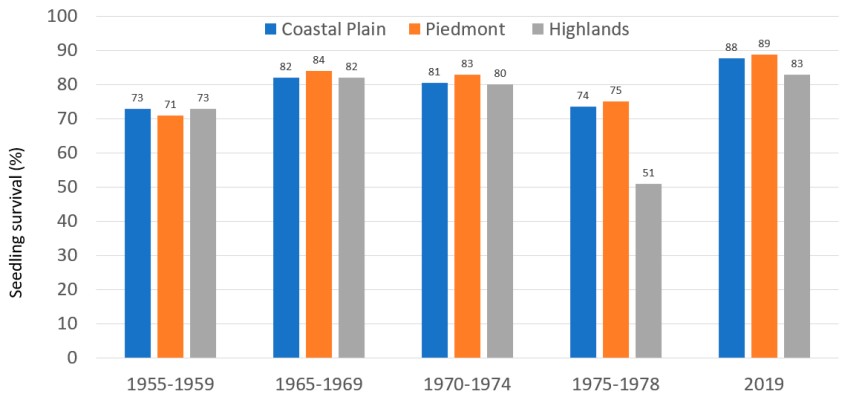

**Figure 1.** Early survival of pine seedlings reported for various periods in the southern United States. Adapted from [4–8]. Coastal Plain, Piedmont and Highland provinces were defined by Schultz [9]. The least significant difference (α = 0.05) for a one-tail test is −9.3% and ±11.4% for a two-tailed test.

Most agree that rainfall plays a major role in the survival of planted seedlings. If average rainfall has increased, this might help explain some of the increase in the rate of survival. In fact, for one region in the SUS, the average rainfall for 2019 to 2021 was about 19% greater than for the period from 1955 to 1959 and 15% greater than the period from 1975 to 1978 (Figure 2). Although annual rainfall was near normal in 2019, reported survival for the SUS was 88%, and about one out of 20 planted areas had survival low enough to be replanted. Average mortality can exceed 15% in some years (Figure 3).

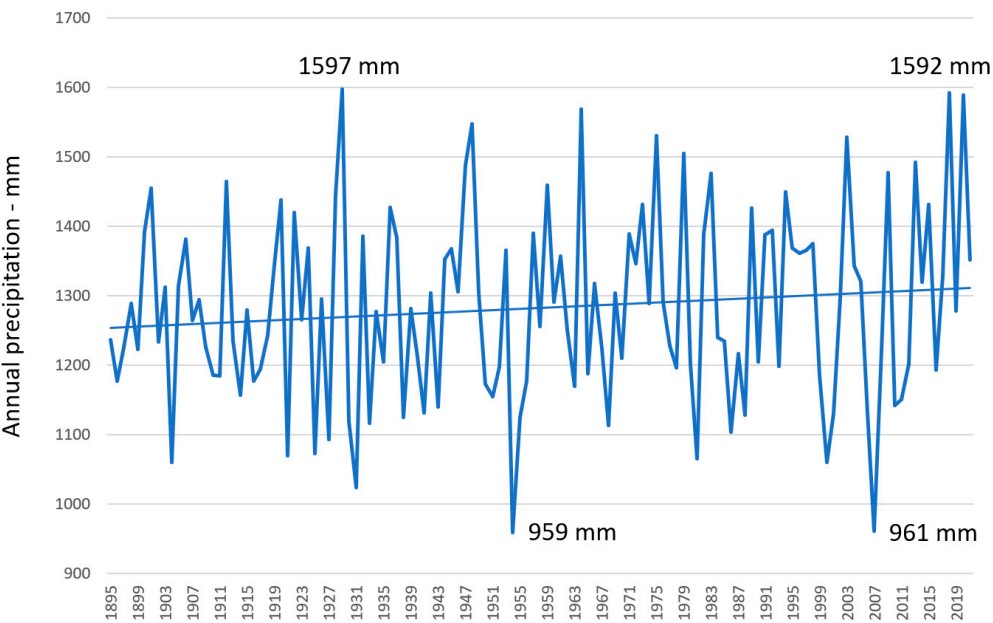

**Figure 2.** Average precipitation (1297 mm for years 2000 to 2021) for the southeast region of the USA (AL, FL, GA, SC, NC, VA) is about 45 mm greater now than 100 years ago. Since 1895, there have been 8 years with rainfall greater than 1500 mm, and 6 of those occurred after 1960.

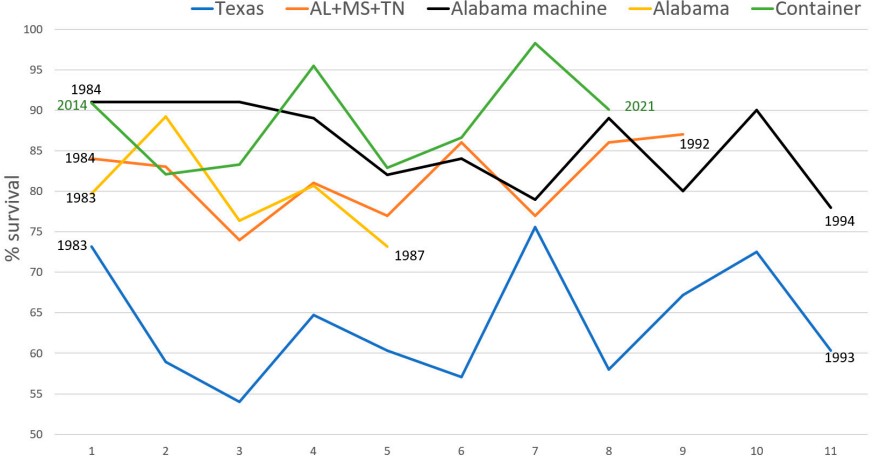

**Figure 3.** Survival of loblolly pine varies with year, region, stock type and planting method. For this graph, survival values before 1995 represent bareroot stock. The green line represents hand-planted container-grown stock, and the black line represents machine-planted bareroot stock. The average rainfall for East Texas (1176 mm) is about 17% less than the average rainfall for North Alabama (1418 mm). Data were provided by Brad Barber (Texas), Steve McKeand (container), Jerry Frame (Alabama machine; AL + MS + TN) and Harry Vanderveer (Alabama).

The objective of this review is to list non-nursery factors that contribute to the initial mortality of pine seedlings. Nursery managers, foresters and landowners may find this list of over 50 factors useful when customers ask what can be done to increase survival. Too

often, landowners want reimbursement for mistakes that were made after seedlings left the nursery. The outline of the paper is as follows. Section 2 presents the literature sources, and Section 3 discusses the importance of keeping good records. Factors affecting survival before planting and during transportation are presented in Sections 4 and 5. Section 6 covers planting, and Section 7 covers beneficial treatments applied after planting. Weather factors are listed in Section 8, and Section 9 involves replanting. Section 10 covers costs, and Section 11 involves use of statistics by researchers.

## 2. Materials and Methods

Since 1975, we have collected a large number of papers published after 1900. The papers were sourced from books, proceedings and repository sites, such as Treesearch, TreeCD, Agricola and Google Scholar. We excluded papers about angiosperms and pseudoreplication and retained papers that contained survival data for pine seedlings grown in nurseries in North America, Canada, Europe and Africa. A few references to the survival of Sitka spruce (*Picea sitchensis* (Bong.) Carr.) and Douglas-fir (*Pseudotsuga menziesii* (Mirbel) Franco) were included. Many papers involved seedling quality, but these were excluded as these effects were adequately discussed in other unsystematic reviews. Papers were retained if they involved factors that killed pine seedlings after they passed the nursery gate. To keep citations to a minimum, a small portion of the total available papers ended up as citations. This review does not address a specific research question, and it does not cite all of the published literature on pine survival. As a result, it is not a systematic review of all literature pertaining to a single question.

Powerpoint software was used to construct graphs to illustrate treatment effects on the survival of planted stock. Among the 23 graphs, about half present survival data for both the *Y*-axis (treatment) and *X*-axis (control). This graphical technique was utilized since it is relatively easy for landowners to understand. It also illustrates that treatment response is often greater when mean survival is less than 80%.

## 3. Record Keeping

When seedling survival is unexpectedly low, landowners need to know why so that mistakes are not repeated in the future. Sometimes the reason for low survival is obvious, but occasionally, consultants are hired to determine the cause. When experts arrive soon after the onset of mortality, dying seedlings can be examined, and the cause may be determined. However, when expert arrival is delayed, important clues begin to dissipate and are gone after seedlings have dried and decomposed. When this happens, accurate records provide needed clues (see planting log in Supplementary Materials).

Unfortunately, key records are often missing. Important information includes lifting date, storage length, method of transportation to the site, seed origin, seedling morphology, planting date, temperature in storage, planting depth, soil moisture when planting, fertilizer placement and rate. We recommend recording the standardized precipitation index at planting (see Supplementary Materials). If an herbicide was applied, the total amount purchased and the total area treated should be recorded along with the application date. Weather records are very important. In 1977, weather stations in the Cumberland Plateau in Tennessee recorded the coldest January since 1895. This likely explains why pine survival averaged 51% in the highland region of the SUS (Figure 1).

Pine seedlings can die within 30 days of planting [10] or death can linger for 6 months to 5 years or more. When mortality is acute (1 to 3 months after planting), the cause can generally be traced to one factor. Heat exposure due to improper handling/storage or debarking weevils are examples of quick mortality. However, more often, chronic mortality occurs within the first five years after planting. The root cause can then be difficult to ascertain as a multitude of factors generally contribute to mortality. For example, machine-planted seedlings receiving cold damage on soils with high available water hold capacity with adequate rainfall might not die. However, cold damaged seedlings can contribute to chronic mortality when combined with factors such as root pruning, planting seedlings with

the root-collar near the surface, nematodes and low soil moisture combining to eventually kill the seedling due to too much transpiration. Mortality analysis needs to be conducted by foresters with extensive reforestation experience to avoid repeating mistakes of the past. Often the blame is placed on the nursery when the actual cause is experienced after the seedlings leave the nursery.

## 4. Preparing the Site

Prior to 1980, site preparation in the SUS was dominated by mechanical methods such as shear rake and pile, bedding and disking. These treatments helped to facilitate machine planting and reduced the growth of hardwood sprouts. Site preparation and road maintenance contractors utilized the smaller crawler-tractors in the winter to pull planting machines, and during the spring and summer, they used the same machines to cultivate the soil and conduct road work. Keeping crews and machines working year-long is a significant economic advantage. Integrated forest products foresters favored mechanical site prep, and many experienced foresters of that day commented that woodland organizations were reluctant to change from mechanical soil cultivation to chemical site preparation for fear of losing a "mechanical empire", for lack of a better description. Since machine planting often results in deeper planted seedlings, average survival from machine planting can be greater than with contract hand-planting crews. After effective chemical weed control methods were adopted, companies realized there was little additional benefit from disturbing the soil surface. As a result, the frequency of soil cultivation declined, and the use of hand-planting increased.

### 4.1. Burning

Each year, prescribed burning occurs in the SUS on about 9% of sites before planting pine seedlings. The goal is to control small hardwoods that compete with newly planted pines and to make hand-planting easier [11]. At a flatwood site in Florida, burning before planting increased survival by 7% [12]. Coosa County, Alabama, was in an extreme drought from July 2007 to March 2008, and seedlings planted in February on burned sites averaged 66% survival, while survival on not burned areas averaged 54% (Figure 4). When seedlings were hand planted in February 2008, burning increased survival by 21%. Possible reasons for this effect include less hardwood competition for moisture, duff in planting holes, fewer live insects in the topsoil and more rainfall reaching the soil. At some locations, burning increases soil moisture [12,13].

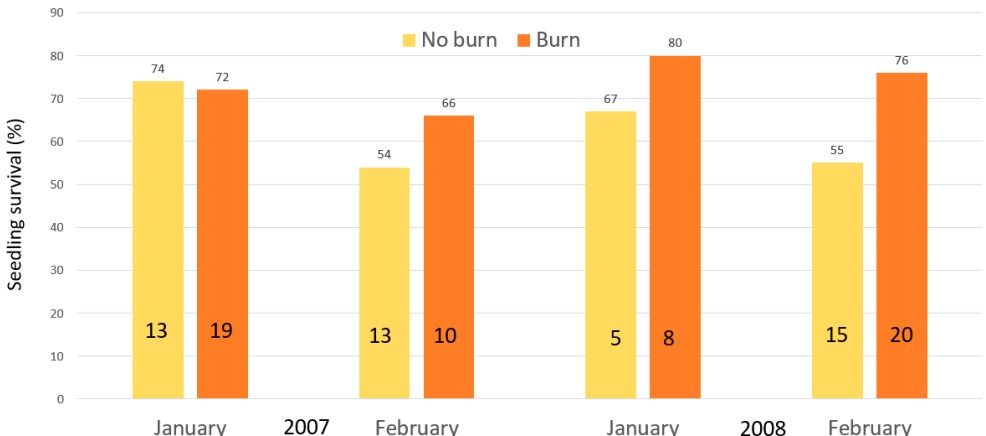

**Figure 4.** Prescribed burning (before planting in January or February) increased the survival of bareroot loblolly pine seedlings during a severe drought in central Alabama (data from Al Lyons; number on bar represents number of sites per mean). The increase in survival was significant except for January 2007. The least significant difference ($\alpha = 0.05$) for a one-tail test is $-11\%$ and $\pm 14\%$ for a two-tailed test.

Site preparation burning in the SUS has declined over the last four decades for various reasons. In 1980, 98% of the area planted to pines in the lower Coastal Plain was burned prior to planting. During the era of integrated forest product companies, woodland organizations had a significant workforce and enough equipment to conduct road construction, maintenance and site preparation (mechanical, burning, etc.). After these companies monetized timberlands, this workforce capacity was lost, which lowered the amount of burning. Only a few contractors had the capacity to conduct burning. There were also environmental (smoke management), liability (smoke, wildfire) and soil nutrient concerns that limited wide-scale burning [14]. Currently, site preparation burning is limited to sites with enough slash and debris to slow hand planting and reduce planting quality or where there is a high density of wildling pines. (Appendix A Figure A1).

*4.2. Bedding*

In the SUS, bedding has been used on sites where high-water tables reduce soil oxygen and reduce pine survival [12,15]. In southeast Georgia, burning and bedding increased survival by 7% (Figure 5). However, at some locations, bedding increased seedling mortality by more than 10% [16]. The additional mortality was attributed to rough, unsettled beds full of air pockets. Therefore, planting too soon after bedding can increase seedling mortality. Bedding quality affects survival as seedlings planted in low dips generally have higher mortality. This is common in single-pass beds, with bed quality increasing with double-pass bedding. Double bedding might increase the survival of slash pine (*Pinus elliottii* Engelm.) by 5 percentage points. In the lower Coastal Plain, about 31% of planted areas are single-bedded while 9% are double-bedded [8].

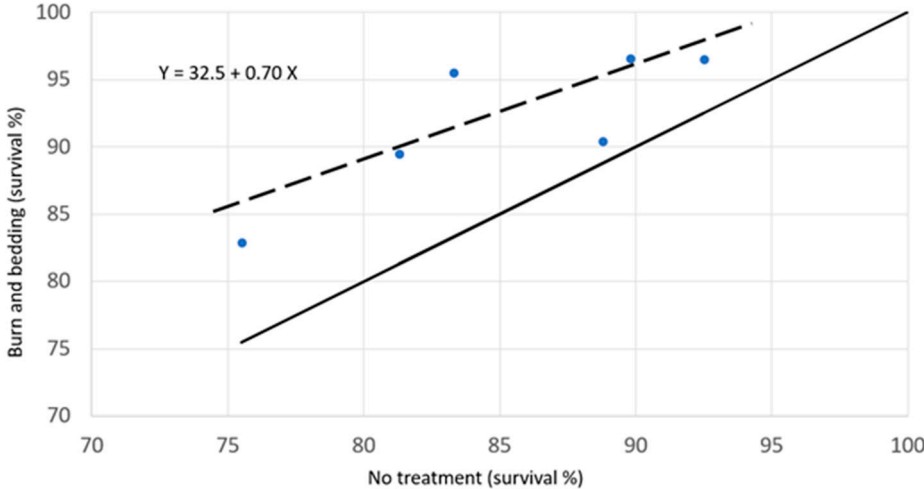

**Figure 5.** Burning followed by bedding increased fourth-year survival of slash pine at six sites in southeastern Georgia [17]. The least significant difference ($\alpha = 0.01$: two-tailed test is 4.5%). Points above the solid line represent sites where burning and bedding increased survival. $p > |t| = 0.19$ for intercept and 0.0464 for slope; $R^2 = 0.67$.

*4.3. Disking*

On some sites, flat disking may increase early volume growth due to a 9% increase in survival, but there may be no increase in volume where disking has no effect on survival [18–20]. Disking is not a common site preparation method today and was most common during integrated forest product management.

*4.4. Subsoiling*

Subsoiling (or ripping) compacted soil before hand-planting can increase survival since seedlings are planted deeper and less foliage is exposed to drying winds. On some sites, ripping soil before planting increased pine survival by 7% or more ([21–24]; Figure 6).

When compared to digging a 20 cm deep hole, ripping to a depth of 40 cm increased the survival of patula pine (*Pinus patula* Schiede ex Schltdl. & Cham.) by 16% [24]. Seedlings should be planted four months after ripping to allow the rain to settle the soil. When the soil in the rip is not yet settled, planting seedlings on the off-set berm will likely improve survival [25]. If the soil has not settled and seedlings are planted in the ripped zone, mortality may result due to reduced root-soil contact or soil erosion burying seedlings [26].

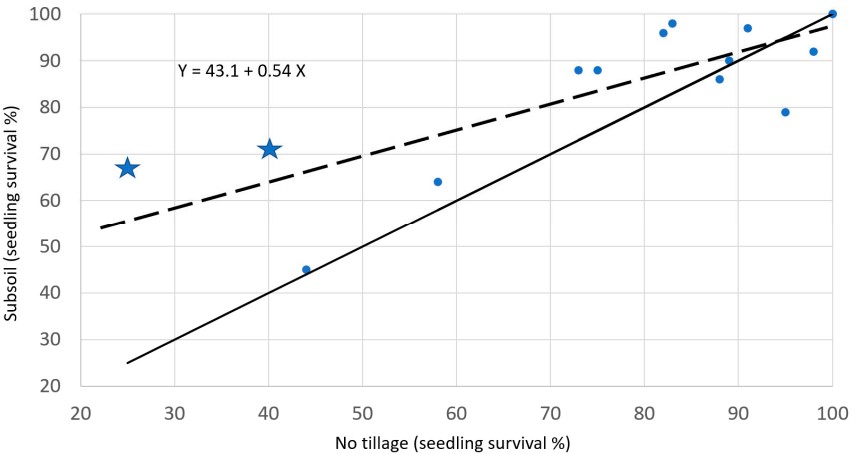

**Figure 6.** Subsoiling increased the survival of loblolly pine seedlings (*n* = 15) by an average of 8 percent [2], and the increase was 13% on sites with less than 90% survival (*n* = 10). Sites with less than 50% survival were planted in early 1995, a year with below-average rainfall. At two sites in Saluda County, South Carolina (Stars), March to May (1995), rainfall was 160 mm below normal. Points above the solid line represent sites where subsoiling increased survival. $p > |t| = 0.001$ for intercept and 0.0025 for slope; $R^2 = 0.58$.

### 4.5. Scalping

Scalping is a recommended practice before planting on former cropland sites. In some cases, scalping increased survival by more than 30% [27–29]. This increase is due to reducing weed competition and reducing the populations of insects and diseases. On sandy soils, scalping might not reduce the rate of water infiltration. However, on fine-textured soils, scalping reduces infiltration and can hold water like a pond. "If the soils are very wet, or the soils are very heavy (high clay content), scalped rows may hold water and drown the seedling" [30].

### 4.6. Liming

Often lime is not needed in bareroot pine nurseries unless the soil pH is below 4.5, and lime is also not operationally applied in North American plantations. In research trials, applying 4480 to 6700 kg ha$^{-1}$ of dolomite did not appear to increase the mortality of loblolly pine [31,32]. However, incorporating a higher rate of 11,200 kg of agricultural-grade lime before planting western white pine (Pinus monticola Dougl. ex D. Don) reduced survival on two sites in 1992 by more than 9% [33].

## 5. From Nursery Gate to Planting Site

Boxes, bags and bales of seedlings are transported to the planting site in open trucks, insulated vans and refrigerated vans [34]. Seedling mortality increases when open trucks-trailers transport seedlings for long distances during −4 °C days. Therefore, transportation should be in refrigerated vans or covered vehicles that are preferably insulated [35,36].

### 5.1. Improper Storage

When soil is moist, it is generally best for landowners to plant seedlings soon after seedlings arrive at the planting site. When weather delays planting, then southern pine

seedlings should be stored in a cool environment to reduce heat, which encourages fungal growth on roots and foliage. In a trial in Tennessee, survival was greater when seedlings were planted soon after delivery than when seedlings were planted after 9 weeks of storage in ambient conditions (Figure 7).

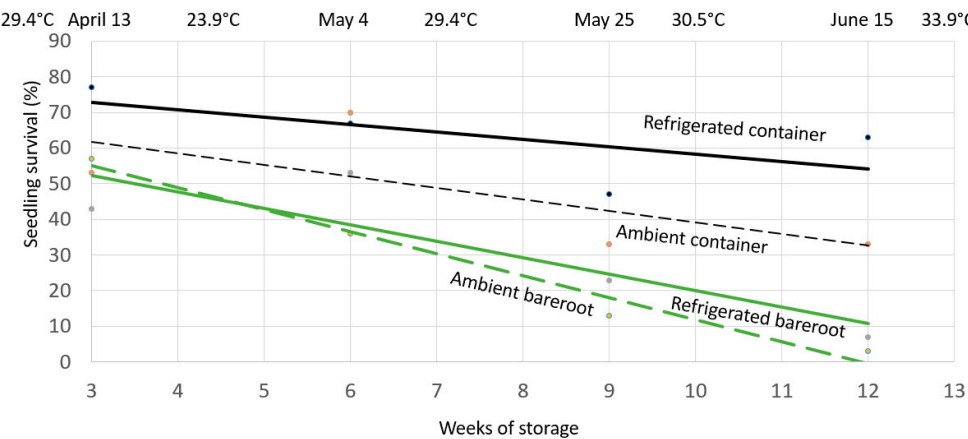

**Figure 7.** Effect of date of planting, storage length and storage method on survival of loblolly pine planted in Scott County, Tennessee [37]. The data presented assume 3 weeks of storage prior to planting on April 13. Storing seedlings reduced survival, as did storing seedlings without refrigeration. The highest air temperature recorded for each 3-week period is indicated at the top of the graph. There were 30 seedlings planted per treatment-date combination. Bareroot seedlings, likely produced at Pinson, Tennessee, and container seedlings (likely Spencer-Lamaire root-trainers) were grown in a greenhouse at the University of Tennessee. The least significant difference is −13.7% ($\alpha = 0.05$: one-tailed test). For ambient bareroot seedlings, $p > |t| = 0.0057$ for intercept and 0.012 for slope; $R^2 = 0.97$. The slopes of the remaining lines were not significantly different from zero ($p = 0.13$ for refrigerated bareroot, 0.30 for ambient container and $p = 0.36$ for refrigerated container).

A few landowners have access to coolers and store seedlings for a few weeks before planting. Some coolers are equipped to keep the humidity high while others act as a dehumidifier and can dry seedlings over time. Spraying water into the cooler on a daily basis will help maintain the humidity within the cooler. If seedlings are not stored properly, they can dry during storage and mortality is increased. Small quantities of seedlings may be stored by heeling trees in sawdust or soil.

In some countries, containers with seedlings are shipped to the planting site, and pines are extracted just prior to planting. When freezing weather is imminent before all seedlings are planted, the remaining containers with seedlings can be placed on the ground and covered with an insulated tarp. This reduces the risk of root injury to unplanted seedlings.

### 5.2. Water in Bags

Landowners should be careful not to puddle water in bags since "excess water can drown root tips or promote mold on the seedlings" [38]. In one trial, adding 2 L of water to bags increased the mortality of bareroot spruce seedlings that were stored for 2 weeks at 20 °C [39]. In another trial with bareroot loblolly pine, seedlings were lifted in January, and 86 mL of water was added per 1000 g of seedlings. Seedlings were stored in kraft-polyethylene (KP) bags for 2 weeks at 2 °C, and this reduced root growth potential by more than 75% [40]. This occurred even though no mold or fungal growth was observed on the roots or foliage.

### 5.3. Improper Transportation

The preferred way to transport large quantities of seedlings is in a refrigerated van [41], and small quantities may be transported in insulated trailers [34]. The temperature should

be recorded to ensure seedlings do not freeze or overheat. Seedlings have been killed when transported in the back of a truck during freezing temperatures.

Sometimes seedling packages are dropped when unloading seedlings at the planting site. Dropping container-grown lodgepole pine 30 times (1 m height) did not reduce seedling survival [42]. However, dropping other conifers may reduce root growth potential and decrease height growth [43–45].

*5.4. Transportation Too Far North*

Genetics plays a role in seedling survival, especially in terms of response to a hard freeze. For example, some southern Coastal Plain sources of loblolly pine do not survive well when planted in southern Illinois, where temperatures might reach −20 °C [46]. Likewise, lower survival is expected when planting Florida sources of pine in Virginia. For example, the survival of a local Virginia source of longleaf pine at age 10 years had 80% survival, while a seed source from Florida had 51% survival [47].

Due to first-hand experience with de-acclimation freezes, several forest companies in the SUS reject flawed computer-based recommendations to plant pines 700 km north of their native location (i.e., location of the mother tree). Most loblolly pine seedlings are planted within 5° latitude of the native origin (≈550 km). For example, in a survey of members of the North Carolina State University Tree Improvement Program, only one member experienced freeze damage when a southern coastal family was planted too far north [48]. No member reported outright failure due to establishing plantations using a single family. Only about 8% of progeny-test plots established with container-grown loblolly pine were abandoned due to survival rates lower than 60%.

## 6. Planting

*6.1. Root Pruning*

Most tree planting guides place too much emphasis on avoiding bent roots and not enough emphasis on keeping roots. The "bent-root-kills" myth has lowered seedling survival because of publications that recommend pruning up to 50% of roots to avoid bent lateral roots and L-shaped taproots. The unjustified fear of a 3-cm bend of the taproot at the bottom of a 20-cm hole encourages (1) removing a portion of the taproot and (2) making a 13-cm deep hole to match a 13-cm pruned taproot. When soil is dry, both practices contribute to an increase in mortality.

In order to make tree planting quicker, one planting guide states to "prune roots to a uniform length by aligning root collars in bunches before pruning" and to prune roots no shorter than 13 cm. In fact, pruning to a 7.6 cm length can reduce pine survival in a dry year by 13 to 28% [49]. These recommendations suggest it would be OK to prune a 20-cm taproot (with six lateral roots) so that after pruning, the taproot would be 13 cm long with four lateral roots remaining. Of course, cutting the tap root will make hand-planting easier and will reduce the frequency of taproots with a 3-cm bend at the end. However, removing fibrous roots lowers root-growth potential [50,51] and increases mortality. For example, pruning 20 cm taproots to a 13 or 16 cm length reduced loblolly pine survival by 4% [52,53] and at one location, pruning to a 7.6 cm length reduced survival by 18%. For this reason, a few tree planting guides say "do not allow planters to prune roots." A contributing reason why machine planting often results in better survival is large roots with long taproots do not slow the rate of planting. As a result, machine planters do not prune roots.

*6.2. Root Stripping*

Some hand planters will strip roots just before they insert the seedling into the hole. This practice (typically performed by moving the root through a closed fist) removes some of the small fibrous roots and ectomycorrhiza. As a result, root mass might be reduced by 2%. However, the ability of the seedling to produce new roots can decrease by more than 40% [51], which can increase mortality by 12% or more [54].

## 6.3. Root Exposure

Drying roots before planting can reduce root growth, which leads to mortality [55]. In one trial, exposing pine roots outdoors for 15 min reduced new root growth by 70% [56] and reduced survival by 100% [57]. Exposures of 5 to 10 min on a sunny day can reduce pine survival by 5% to 18% [58–61], and 30 min reduced survival by 10% to 35% (Figure 8). To increase the probability of survival, planting stock should be protected from direct sun and kept cool. Planter handling and transport should minimize seedling exposure to prevent air drying [35,56].

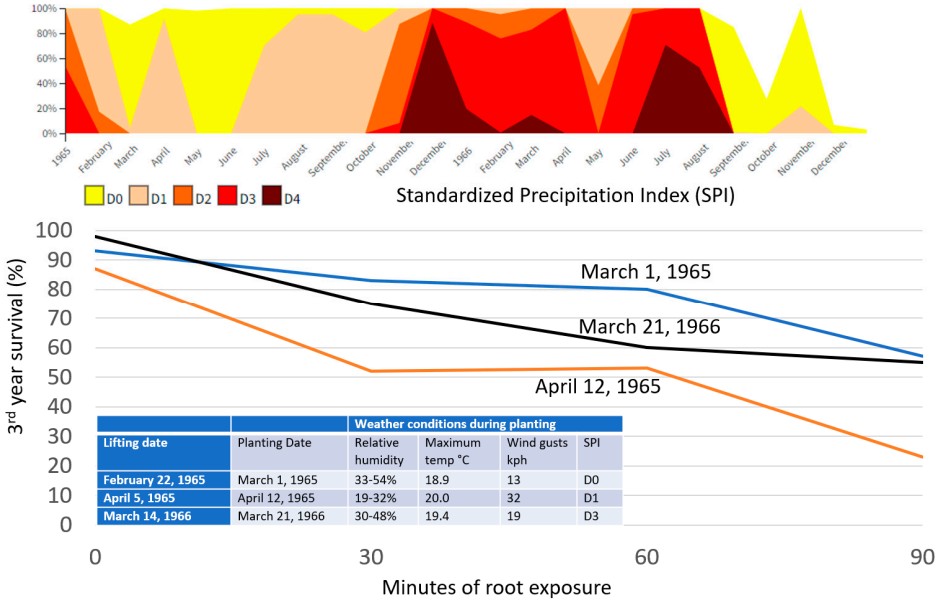

**Figure 8.** Exposing roots to 30 min of sun in Virginia [62] and then planting in moist soil (Standardized Precipitation Index; SPI = D0; February rain in Appomattox County = 94 mm) reduced survival by 10%. In contrast, when planted in drier soil (SPI = D1; April 1965 rain = 53 mm), the same exposure reduced survival by 35%. Likewise, planting 30-min exposed seedlings in dry soil the following year (SPI = D3; March 1966 rain = 52 mm) reduced survival by 23%. Data from [62]. The least significant difference ($\alpha$ = 0.05) for a one-tail test is −10% and ±12% for a two-tailed test.

## 6.4. Machine Planting

Pine seedlings are planted either by machine or by hand. From a 2020 survey of 376,000 ha in the SUS, 40% were planted by machine (Figure 9). Hand planting is often used on rough sites in the Piedmont and, when available, machine planting is preferred on flat lands in the Coastal Plain. When availability and the costs are reasonable, most regeneration foresters in the SUS prescribe machine planting since the probability of good survival is greater than that for hand planting bareroot stock. In Florida, 30 out of 64 sites were planted with machines [63].

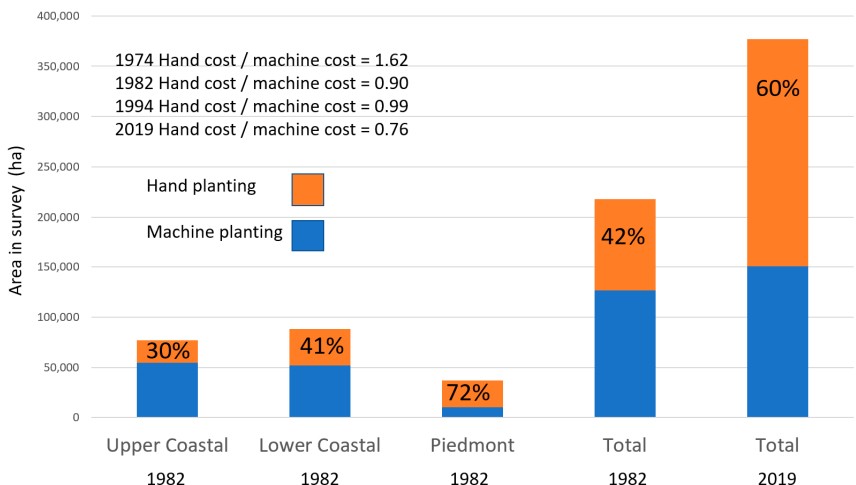

**Figure 9.** As the relative cost of machine planting increases, the proportion of hectares planted by hand increases. Data from [8,64].

In Louisiana (December 2000 to March 2001), machine planting on 11,220 ha averaged 87% survival, while hand planting (34,564 ha) averaged 80% survival. For one company in Alabama, machine planting bareroot pine averaged 86% survival, while hand planting averaged 75% survival (Figure 10). The increase in survival was mostly due to placing roots deeper in the soil [65] because machines do not get tired of making deep holes at the end of the day and because hand planters were allowed to prune roots. In years when rainfall was below average, machine planting resulted in 16% greater survival than hand planting. In contrast, when (January-June) rainfall exceeded 800 mm, survival with machine planting was only 5% better than hand planting. Some machines do not plant container seedlings as well as hand planters [66]. In fact, some are reluctant to plant container-grown longleaf pine with machines since the risk of burying the plug is great [67]. In one trial, machine planting of container-grown stock resulted in 64% and 70% survival in scalped and non-scalped areas, respectively [68].

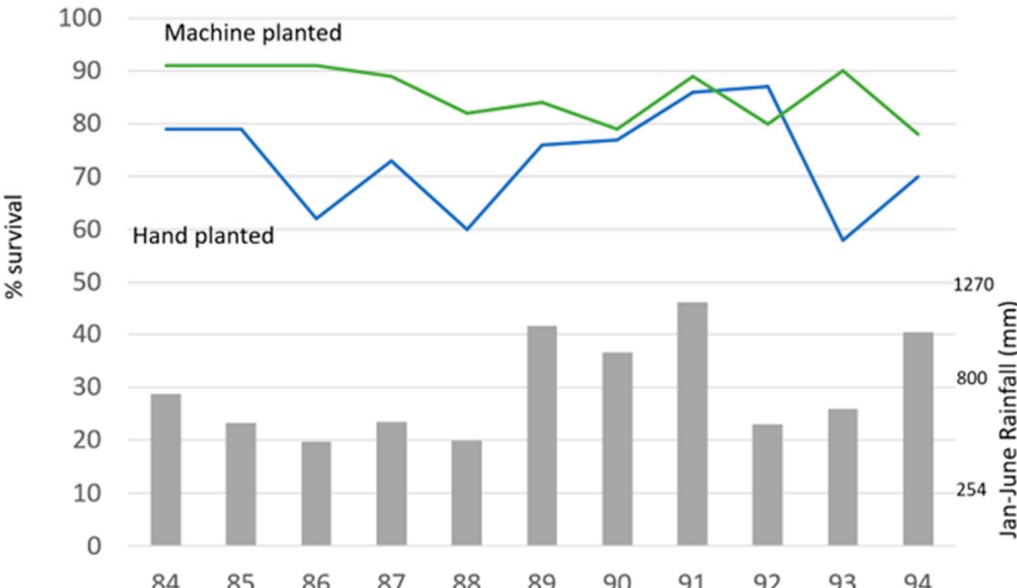

**Figure 10.** Survival of bareroot loblolly pine in northeast Alabama varied with rainfall and planting method. Over the 11-year period, early survival averaged 86% and 75% for machine and hand planting, respectively. The least significant difference ($\alpha = 0.05$: one-tail test is $-6.4\%$). Data from [69].

Various brands of machines are used to plant seedlings [70,71], and some machines can plant 400 to 800 seedlings an hour, regardless of stock type. In 1982, the time required to plant 10 ha of upper coastal plain soil was 11 h for a machine-planting crew or 15 h for a hand-planter [64]. In contrast, some container-only machines plant less than 400 seedlings per hour [72,73]. The time required to machine plant 1000 pine seedlings depends on the spacing between seedling rows. In theory, a spacing of 5 m between rows would require half the time as a typical 2.5 m row spacing, which would lower the cost of machine planting.

The cost of planting 1000 seedlings could be $120 with hand planters or $170 with machines. If average survival is greater with machine planting, the $50 greater cost might increase survival by 0% for wet years and 20% during dry years (Figure 10). When planting 1000 seedlings per ha, a 10% increase in survival equates to an additional 100 living seedlings. When a seedling costs 20 cents and planting costs 12 cents, 100 dead seedlings would equal a loss of $32. Many landowners choose hand-planting and are unwilling to spend $50 in order to save $32 in dry years. However, at a cost of 50 cents per seedling, a $50 investment in machine planting might save $62.

### 6.5. Hand Planting Tools

Many types of tools are available for hand planting but generally, the size of the root system is a factor when selecting tools for operational use. A metal pipe used to plant a 30 cm$^3$ container plug is not suitable for planting a seedling that was grown in a 1000 cm$^3$ container. A planting spade is the preferred tool in New Zealand, while a shovel is popular in Oregon [74]. In the SUS, bareroot seedlings are typically planted with either a planting bar or hoedad [75]. All four tools can be used to make a 25 cm deep hole. When the hole is deep enough, soil moisture is adequate, and the planting method is correct, good survival can be achieved. Tools designed to make a 10 cm deep hole are not recommended for planting bareroot seedlings.

### 6.6. Family Block Planting

Planting advanced, half-sib and full-sibling families of loblolly pine in family blocks is a common practice in the SUS. In 2019, over 45% of loblolly pine stands were established using second-cycle seedlings [8]. In theory, the method of planting (family-block vs. mixed genotypes) does not affect overall seedling survival during the first year. However, some foresters notice lower than expected survival when improved genotypes are planted in single-family blocks. Due to record keeping, one organization noted a certain family had 10% lower survival than others [48]. Large landowners can use this knowledge to their advantage when planting half-sibling blocks the following year.

In contrast, some landowners own small forests, and they might plant seedlings once every 25 years. If they decide to plant a mixture of 30 genotypes, they might achieve 96% survival in a good rainfall year or 56% survival in a droughty year (Figure 11). However, if they are unlucky and plant a fast-growing family with the lowest survival, they may get only 29% survival during the bad year.

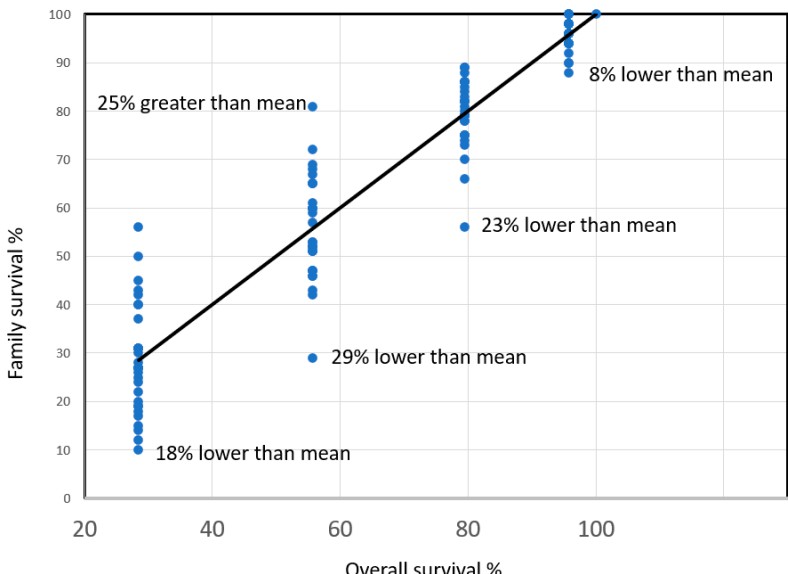

**Figure 11.** When planted in a good environment, the range in seedling survival among 30 loblolly pine families (100% to 88%) is relatively low [76]. As site conditions worsen, it becomes easier to identify families with low potential for survival when records are maintained and reviewed. Data from [76].

### 6.7. Seed Source

There is significant genetic variation within the natural range of loblolly pine, extending from southern New Jersey to southeast Texas. Limited to the north by cold temperatures and to the west by rainfall, native loblolly pine races have developed that are adapted to local climates. The adaption to local climatic conditions is referred to as geographic variation. Seeds harvested from these geographic areas vary in their potential for adaptation and survival depending on where they are planted. Seeds sourced from warmer climates within the loblolly range grow faster than those from more northerly climates. However, the principal factor influencing survival within natural ranges is the average yearly minimum temperature at the seed source's native location [77]. Moving warmer climate loblolly pine seeds far north can result in poor survival [46] and long-term adaptability issues due to cold, snow and ice damage. Eastern loblolly pine seed sources, such as Atlantic coastal material, grow faster than western sources. The slower-growing seed sources of southeast Texas are more tolerant of drought, as they have adapted to lower rainfall and drought conditions. These western sources of loblolly pine cease growth immediately at the onset of drought, whereas seed sources from eastern geographic regions tend to continue to grow. Movement of Atlantic coastal material into the western Gulf coast region is common, which could reduce survival during drought conditions.

### 6.8. Root Soaking

Ideally, roots should not be allowed to dry in storage or during transportation. Since this does occur [78–80], soaking roots in water might increase the probability of seedling survival ([44,79,81]; Figure 12). In Pennsylvania [80], loblolly pine roots were soaked for 24 h before planting, and nine more seedlings survived when compared to nonsoaked seedlings (40 seedlings planted per treatment). However, soaking roots for 4 h before planting reduced survival in Mississippi for some unknown reason [81].

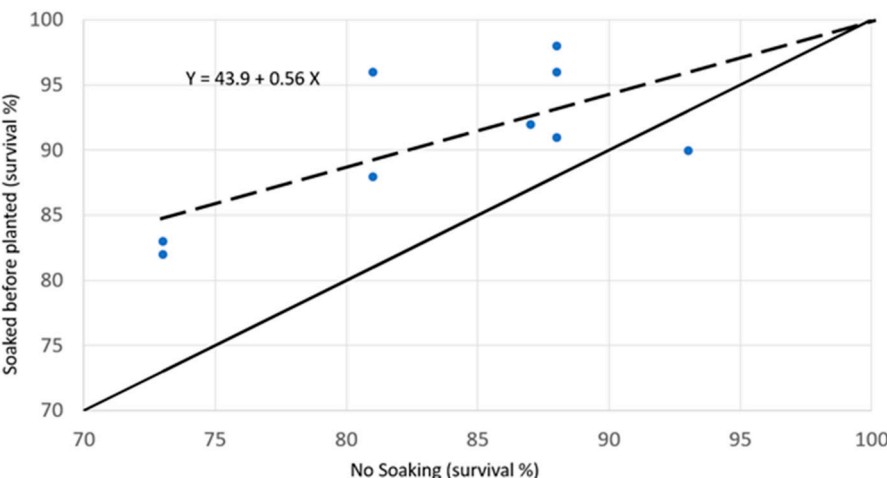

**Figure 12.** Loblolly pine seedlings were lifted and packed into bundles in January at the New Kent Nursery in Virginia (1979–1981). Seedlings were stored in an unheated garage for periods of 4 to 9 weeks. Just before planting, selected bundles were soaked in water for 1 h and then outplanted. Seedlings removed from other bundles were planted without any soaking. On average, soaking seedlings prior to planting increased survival by 7% [82]. Points above the solid line represent sites where soaking roots increased survival. Data from [82]. $p > |t| = 0.046$ for intercept and 0.036 for slope; $R^2 = 0.48$.

### 6.9. Washing Roots

Soil often adheres to the roots of bareroot seedlings, so some landowners may be tempted to wash pine roots before planting. Although soaking roots for a few minutes can be beneficial when roots have been desiccated, washing roots is not recommended since it can reduce survival. Six decades ago, Thomas Swofford said, "When you are washing your seedling roots with water you are also destroying some of your mycorrhiza, as well as some of your rootlets" [83]. In one trial, washing loblolly pine roots reduced survival by 13% [84].

### 6.10. Root Coatings

Several moisture-holding materials have been applied to seedling roots to protect them from desiccation prior to outplanting. These include sphagnum moss, kaolin clay and hydrophilic polymers, also known as polyacrylamide gels. Polyacrylamide gels began to appear in the late 1960′s [85]. Today, pine roots in the SUS are treated with polyacrylamide gels to (1) protect seedlings from desiccation or (2) to improve water relations after planting. The amount of water mixed with 1 kg of polyacrylamide can vary from 300 to 1100 kg. The amount of gel used per 1000 seedlings can range from 2 to 6 kg. Using the right type of gel can increase survival (Figure 13), but the wrong type can decrease survival after planting [86–88]. A clay dip was popular in the past, but a gel treatment provides better protection, costs less, is less messy and requires less transportation and storage space. This explains why only three nurseries treated roots with a clay dip in 2022. The preference for a clay dip is not based upon research but instead is based on tradition, and because the clay is readily visible on the roots. When nurseries first switched from clay to gels, landowners complained they could not see the gel on the roots.

When environmental conditions are favorable, and roots have been protected, there are no expected gains from applying gel to roots before outplanting. However, when conditions are not favorable, gel-treated roots can reduce mortality [87]. When seedlings treated with hydrogels were exposed for 2 h before planting, survival was 40% higher than roots treated with roots dipped in water. Similarly, in 8 of 20 studies, applying a gel might increase survival by 40% [89].

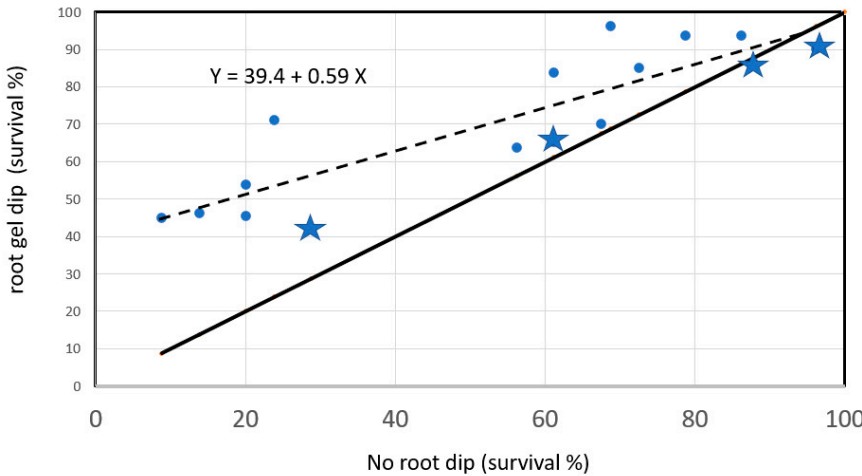

**Figure 13.** Effect of applying a gel root-dip on survival of bareroot red pine seedlings (*Pinus resinosa* Ait.) and jack pine (*Pinus banksiana* Lamb.) [88]. Seedlings not exposed to the sun are represented by stars, while the remaining dots were exposed to the sun for 5, 10 or 20 min prior to planting. Points above the solid line represent sites where the root gel increased survival. *p* > |t| = 0.0001 for intercept and 0.0001 for slope; $R^2$ = 0.77.

### 6.11. Clipping Foliage

Tall bareroot pine seedlings with too much foliage have an increased risk of mortality when planted on dry sites or in years with limited rainfall [90–94]. How much foliage is too much for a pine seedling depends on the amount of root mass, species, stock type and date of lifting from the nursery. A landowner has three options when delivered seedlings do not meet the target seedling's shoot/root mass ratio. Ideally, the nursery will exchange seedlings for stock that is closer to the desired target seedling. When this is not possible, pine seedlings with tall shoots can be top-pruned to the desired height or can be planted with the root-collar 10 to 15 cm below the surface. Planting seedlings deeper on well-drained sites reduces the amount of transpiring foliage.

For bareroot longleaf pine, clipping needles (e.g., length about 15 cm after clipping) just before planting reduces transpiration and typically increases survival (Figure 14). In one study, clipping needles reduced transpiration by 30% and increased survival by 10% or more [92]. However, clipping needles down to 2.5 cm is too short, and this will reduce survival [93,94].

### 6.12. Planting Hole Depth

There are two schools of thought regarding planting hole depth. One school says to make a hole the same depth as the taproot, while the other school believes in making a deeper hole using a machine or shovel. In well-drained soils, a 20 cm deep planting hole will increase survival (of stock with a 30 cm shoot) when compared to a 12 cm deep hole. This is because placing roots closer to a moist soil zone will increase survival. When sufficient rainfall results in high survival, planting 30 cm tall pines deep with 15 cm above ground will increase survival. In some cases, the height of pine seedlings is only 10 cm long, and a deep hole is not necessary for such small seedlings. Planting small container-grown pine seedlings with the root-collar 5 to 6 cm deep is recommended in Finland [95], and an 8 cm depth can reduce damage from insects and drought [96].

### 6.13. Container Plug Exposed

There are two schools of thought regarding planting depth for container-grown longleaf pine (*Pinus palustris* Mill.) Some say the top of the plug should be completely covered with soil to prevent "wicking" moisture out of the plug [38,97–100]. This idea might have originated at the Wind River Nursery, where containers (diameter 6.4 cm; height 25 cm) were fabricated using paper towels. At the planting site, the paper containers were planted

in holes about 28 cm deep. Betts [100] said, "If the toweling sticks above the ground, it will act as a wick and pull the moisture from the area around the tree roots to the soil surface where it is lost by evaporation. A handful of dirt, duff or other debris scattered over the top of the tube can help seal off the planted tube from loss of moisture to the atmosphere." This extra work would not have been needed if the recommendation at that time had been to place the bottom of the container 30 cm below the surface instead of placing the root-collar at the level of the soil surface. A few years later, regeneration foresters were told to plant container plugs deeper and cover the plug with soil to prevent drying [98,99]. Planting container-grown pine seedlings with root-collar 10 cm below the surface is a valid practice for pines other than longleaf pine. For 30 cm tall seedlings, this practice reduces the transpiration rate and, in dry periods, will increase seedling survival in well-drained soils [101,102].

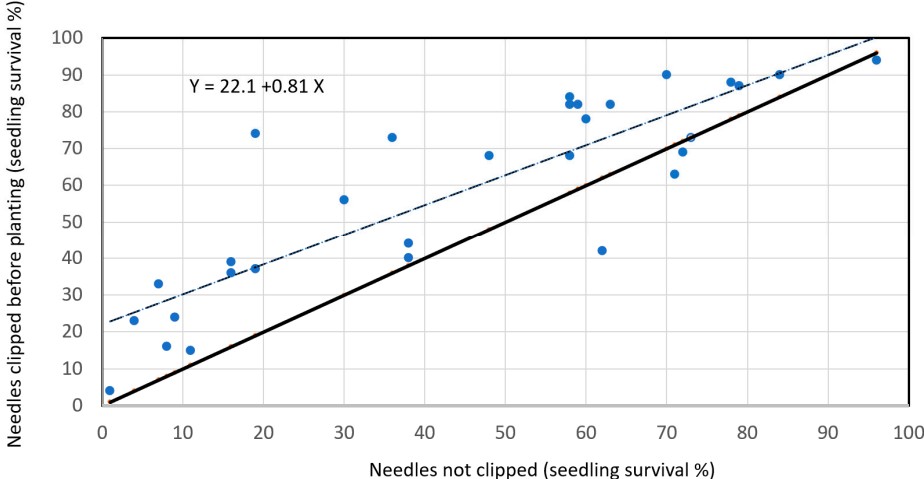

**Figure 14.** Comparison in survival between clipped and nonclipped longleaf pine seedlings [94]. The dashed line represents the regression equation (*n* = 30), and the solid line represents equal survival of the two treatments. Points above the solid line represent cases where clipping needles increased survival. Data from [94]. *p* > |t| = 0.0001 for intercept and 0.0001 for slope; $R^2$ = 0.74.

In contrast, the other school recommends bareroot [103] and container-grown longleaf pine seedlings should be planted with the root-collar 1 to 3 cm above the soil surface. Studies indicate that exposing the top 2.5 cm of the plug did not reduce initial survival [104,105]. In fact, covering the plug with 1 cm of soil increased mortality in Alabama [104]. Although numerous trials with other species show a survival benefit for deeper planting of bareroot stock [102,106,107], there are no data to support the belief that planting longleaf pine seedlings with 1 cm of soil over the plug will reduce mortality. In fact, many bareroot and container-grown longleaf seedlings have died because ponded water covered the terminal bud or because erosion covered the terminal bud with soil.

*6.14. Planting in Wrong Month*

There are two schools of thought regarding the best time to plant container-grown seedlings in the SUS. One group recommends planting in moist soil from October to December 1, while the other school says fully developed seedlings can be planted in any month of the year. Although researchers can sow seeds in any month, operational nurseries typically sow seeds in April or May, and seedlings are ready to plant in October [108]. Planting seedlings before December allows for new root growth, and established seedlings are typically tolerant of root-inhibiting herbicides applied the following May. Many researchers plant longleaf pine between 1 November and 1 February (Figure 15).

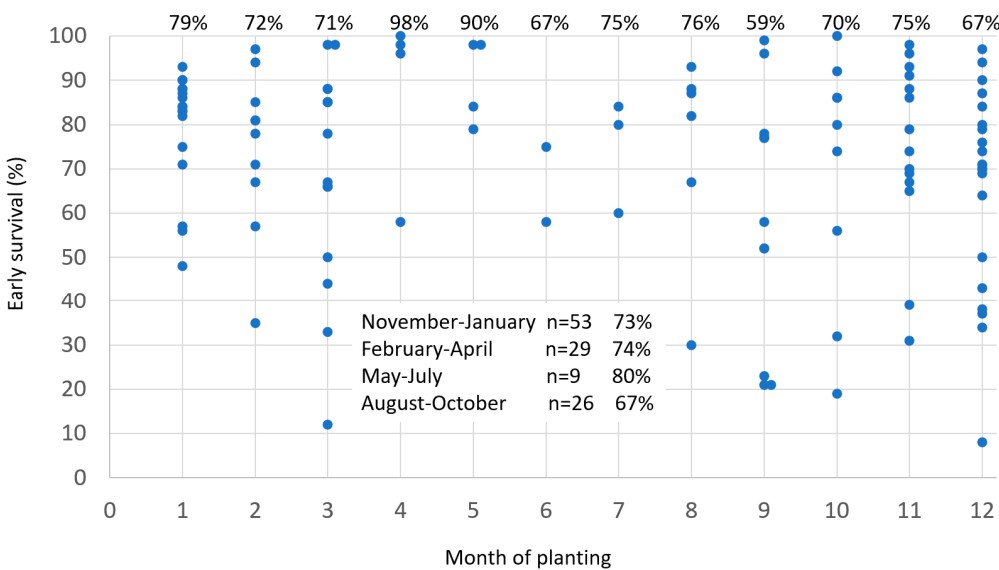

**Figure 15.** Some researchers prefer to plant container-grown longleaf pine seedlings in moist soil during the fall before December 1, but some wait till the end of December when students are not in school. Sometimes low survival in December is related to planting seedlings with an insufficient amount of freeze tolerance [109]. Each dot (*n* = 117) represents one value from a published research trial. For this graph, median survival is 79%, and 24% of the dots are below the 60% line.

Technically, there is no wrong month to plant pine seedlings, only a wrong time to plant. For bareroot pines, the planting season in the deep SUS can extend from October to February, but a dry February can be the wrong time to plant pines (Figure 15). An estimate of the amount planted by month might be 1% in October, 5% in November, 15% in December, 40% in January, 25% in February, 14% in March and <1% in April. In Virginia, bareroot seedlings had 81% to 94% survival when planted before 21 December (Table 1). In Louisiana, March and April are the worst months for achieving high survival with bareroot stock [110], while these months are acceptable in regions north of Tennessee and North Carolina [111,112].

**Table 1.** Survival of bareroot and container-grown loblolly pine seedlings after exposure to dark chilling in a cooler or outdoor chilling (natural chilling) in seedbeds at the Garland Gray Nursery in Virginia. Data from [112]. Rainfall and freeze data from Appomattox, VA. Dark chilling does not increase the freeze tolerance of loblolly pine seedlings [109], and dark freezer storage does not increase the freeze tolerance of ponderosa pine (*Pinus ponderosa* Douglas) seedlings [113]. The chilling hours listed below are hours between 0 and 7.2 °C.

| Stock Type | Packing Date 2018 | Planting Date 2018–2019 | Rain during Month mm | Dark Chilling Hours | Natural Chilling Hours | Survival % | Freeze Event 2018–2019 |
|---|---|---|---|---|---|---|---|
| Bareroot | 15 October | 15 October | 99 | <8 | 0 | 81 | 11 November −5.5 °C |
| Bareroot | 15 November | 15 November | 179 | <8 | 158 | 86 | 29 November −5.5 °C |
| Bareroot | 18 December | 18 December | 173 | <8 | 549 | 94 | 27 December −3.9 °C |
| == | January | Frozen soil | 86 | == | == | == | 21 January −12.2 °C |
| Container | 15 October | 15 October | 99 | <8 | 0 | 100 | 11 November −5.5 °C |
| Container | 15 October | 15 November | 179 | 744 | 0 | 86 | 23 November −3.3 °C |
| Container | 15 October | 18 December | 173 | 1464 | 0 | 43 | 27 December −3.9 °C |
| Container | 15 October | 15–16 April | 133 | 4368 | 0 | 96 | None |

March and April plantings should be avoided in the SUS since seedlings do not have enough time to establish roots before the hot-dry season. For example, planting loblolly pine seedlings on 1st March in Louisiana had 64% survival [114], while April plantings had 38% and 43% survival in Tennessee and Kentucky, respectively [115,116]. Landowners who

plant in moist soil in October or November are planting prior to the cool-wet season while those who plant in March (below 37° N) are planting just prior to the hot (possibly dry) season.

Container-grown seedlings lifted on 13 November 2001, stored for two months, and planted on 22 January (after freezing temperatures) survived well. In contrast, seedlings stored for 10 weeks and then planted on 22 December with little natural chilling had 38% survival after a −7.2 °C freeze (5 January) (Figure 16). In contrast, a −1.7 °C event on January 26 likely did not injure seedlings planted four days earlier. Mortality due to freezes in December is referred to as the "December dip" [117].

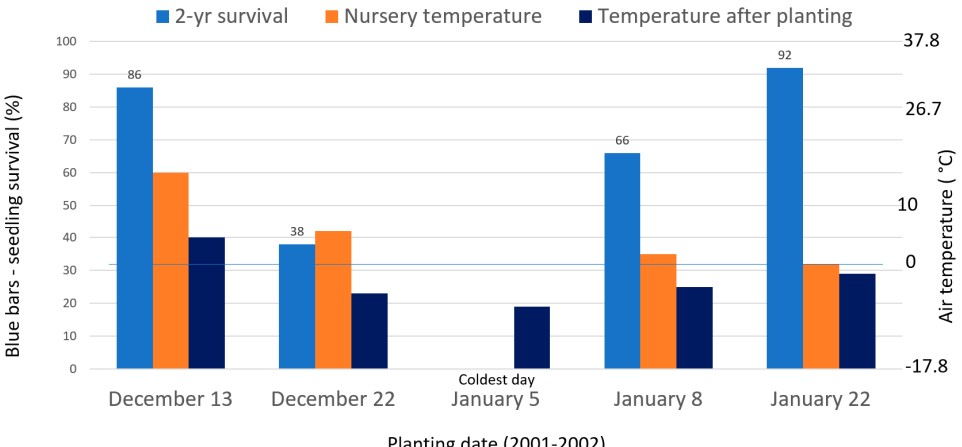

**Figure 16.** The effect of planting date on survival of container-grown longleaf pine seedlings stored for 10 weeks in a nursery cooler at +1.7 °C. Data from [118]. Seedlings were packed in boxes and placed into storage on 2, 16, 30 October and 13 November 2001. The lowest temperature at the nursery (for the lifting date) is represented by orange bars while black bars represent low temperatures recorded at Lumberton, North Carolina, a few days after planting. Accumulated natural chilling hours (0 to 7.2 °C) for Goldsboro, NC was 125 h for 13 November 2019. A temperature of −7.2 °C on 5 January likely contributed to the mortality of seedlings planted two weeks earlier.

*6.15. Adding Water*

When the soil is dry, some landowners add water at the time of planting. In South Africa, container-grown pine seedlings are planted during the summer rainfall season, and 1 or 2 L of water is sometimes added to the planting hole (Figure 17). Some mechanized tree planters are equipped with water tanks to water seedlings as they are planted [119]. Although uncommon, large-scale irrigation systems are sometimes used to establish trees in areas with limited rain. In one trial, the survival of irrigated and non-irrigated pines was 100% and 30%, respectively [120].

*6.16. Loose Planting*

Except for incorrect planting depth, loose planting may be the most important cause of seedling mortality [10,121]. The test for loose planting involves holding two or three fascicled needles between the thumb and finger and pulling upwards. If the seedling is pulled out of the hole, the seedling was too loose, and there was insufficient contact between the soil and roots [121].

*6.17. Pulling Seedlings Up*

Some outdated planting guides say that a "curled root will kill the seedling." Using this myth, a 1989 hand-planting planting guide says to push the roots "deep into the planting hole. Pull the seedling back to the correct planting depth." This method is not used by machine planters who plant the root-collar about 14 to 17 cm below ground [65,107]. If hand planters take the extra time to use the "pull-up" technique, the root-collar will be near the ground line, and they will get paid less due to planting fewer trees. However,

there are no studies to show that when planted deeply, a curled root will kill the seedlings. Therefore, the "pull-up" method may move the roots away from moist soil, expose more foliage, and in dry months, this can reduce survival. Higher survival from machine planting (which typically plants roots in an L-shape) may be, in part, due to not using the "pull up" method. In Canada, planting the root-collar 7.6 cm below the surface increased survival when compared to conventional planting with the root-collar near the soil surface [122].

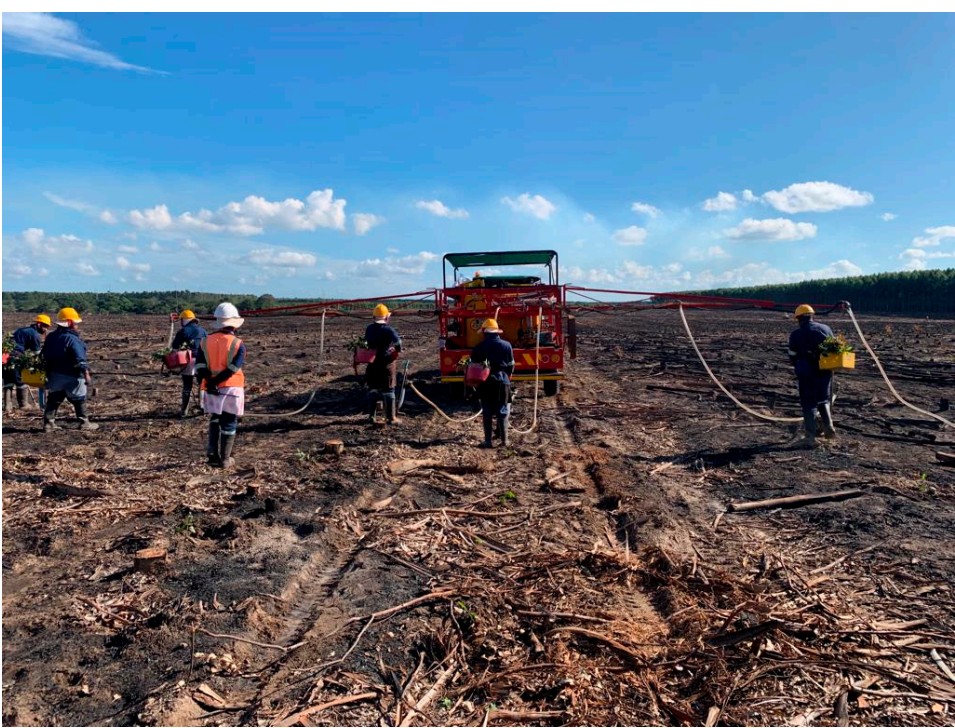

**Figure 17.** In some areas in KwaZulu-Natal, container-grown seedlings are planted during the rainy season (summer). Sometimes 1 to 5 L of water is applied in the planting hole before inserting a seedling. "The use of and/or method of application of the water varies widely between regions and commercial forestry companies. Some companies have a 'no water' policy, others always plant with water, while some only apply water when planting conditions are considered poor (hot weather, dry conditions)" [10]. Applying 2 L of water to young, 7-cm tall, container-grown patula pine seedlings increased survival by 12% at one location but decreased survival of 14-cm tall seedlings by 6% at another site. Adding water had no significant effect at locations where the 100-day survival of non-watered pines was >95%. Photo permission given by Ullrich Hechter—Mondi Forests.

*6.18. Adding Fertilizer*

There are two ways improper fertilization kills seedlings. When weed growth is stimulated, the extra competition reduces soil moisture, and sometimes weeds overtop seedlings. For example, on a site in Tennessee, applying 114 g of N-P-K fertilizer on the soil surface around each seedling increased weed growth, and grasses overtopped seedlings. As a result, the survival of loblolly pine was reduced by 38% [123]. In Texas, a broadcast treatment (140 kg ha$^{-1}$ of diammonium phosphate) reduced survival by 12% at one location and 27% at another [124]. In Alabama and Virginia, a broadcast treatment (280 kg ha$^{-1}$ of diammonium phosphate) did not reduce the survival of loblolly pine [19].

When fertilizer is placed on roots in the planting hole, the salt effect can kill roots. In Louisiana, an application of 0.6 g of fertilizer per seedling killed 66% to 99% of the treated pines [110]. In California, placing 30 g of fertilizer in the hole reduced pine survival by 24% [125]. Likewise, all container-grown loblolly pine seedlings died when a landowner applied an unknown amount of fertilizer directly into the planting hole just before planting. Even when fertilizer (100 g containing 12 g N) was placed in an adjacent hole, 15 cm distant from the seedling, survival was reduced by 12% [126]. To reduce the chance of increasing

weed growth, some researchers place a low rate of fertilizer below ground at the time of planting or several months later. For example, placing a slow-release fertilizer pellet (4.2 g N; $0.15 per pellet) 10–13 cm distant from the roots did not reduce pine survival [127,128].

### 6.19. Adding Fungicide

At some locations, treating the roots of longleaf pine with benomyl and clay just before planting increased survival by 31% [129]. Likewise, applying benomyl in the planting hole increased the survival of patula pine seedlings by 11% in South Africa [130].

### 6.20. Adding Insecticide

In Sweden, about 1% of seedlings died when treated with an insecticide, while mortality from *Hylobius abietis* exceeded 50% on control plots [131]. In South Africa, *Hylastes angustatus* and white grubs were responsible for insect-related mortality of patula pine (Figure 18). Applying an insecticide to the planting hole does not increase survival when insect populations are low, but it might improve survival when nematode populations are high. Typically, most researchers assume nematodes do not reduce the survival of pine seedlings.

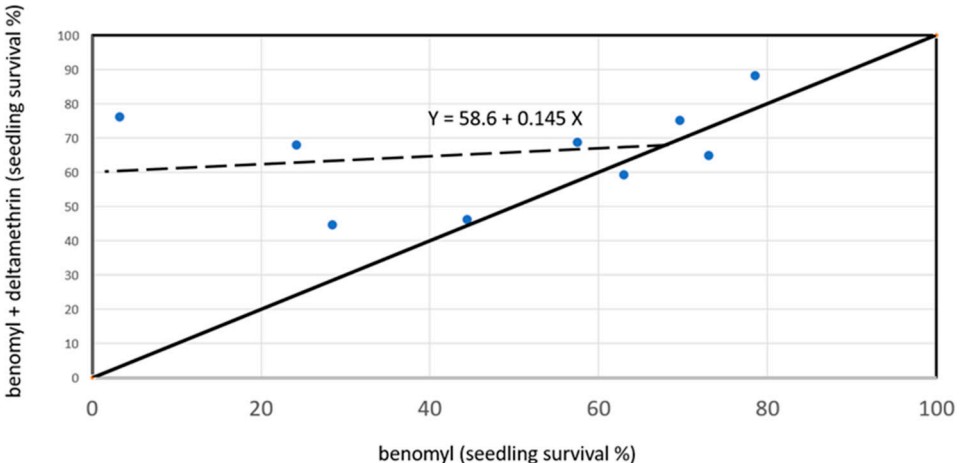

**Figure 18.** Survival of container-grown patula pine seedlings increased when treated with the insecticide (deltamethrin) at Mpumalanga, South Africa. Data from [130]. The first-year survival was 36%, 49% and 66% for untreated, benomyl only and benomyl + deltamethrin treated seedlings, respectively. White grubs and *Hylastes angustatus* caused most of the insect-related mortality, and *Fusarium circinatum* caused most of the disease-related mortality. Points above the solid line represent sites where the insecticide increased survival. $p > |t| = 0.001$ for intercept and 0.49 for slope; $R^2 = 0.07$.

### 6.21. Adding Peat Wedge

When rainfall is not optimal, placing a peat wedge at the bottom of a planting hole can increase the survival of bareroot stock planted on shallow soils. Adding one saturated peat wedge increased the survival of 2-0 pine seedlings by 40% or more [122]. Nearly all container seedlings in the SUS are planted with roots already encased in a peat-based plug (cone or rectangle).

### 6.22. Antitranspirant

Applying an antitranspirant to seedlings before planting will have little effect on survival when soil moisture or rainfall results in more than 85% survival. In Georgia, a 20% increase in survival resulted when a water-emulsifiable organic material (di-1-p-menthene) was applied to pine seedlings in March (Figure 19). This material did not increase survival when seedlings were planted in February (>85% survival) but increased survival by 26% when seedlings were under stress from being planted late in mid-March [132]. Various antitranspirants, are available and some are more effective than others [133].

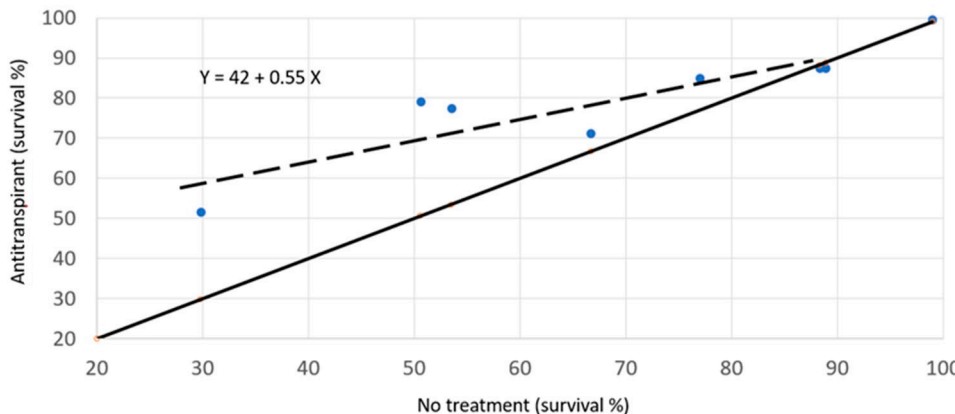

**Figure 19.** Survival of bareroot pine seedlings after treatment with an antitranspirant (di-1-p-menthene). Data from [132]. Seedlings with >87% survival were planted in February, and the others were planted in March. Points above the solid line represent cases where the antitranspirant increased survival. $p >$ |t| = 0.0015 for intercept and 0.0019 for slope; $R^2$ = 0.82.

## 7. After Planting

### 7.1. Vegetation Control

In general, competing vegetation reduces the survival of shade-intolerant pines. Therefore, reducing the biomass of weeds can increase survival in areas where moisture is limiting. In some cases, improper use of application equipment and incorrect herbicide calibration has increased seedling mortality. When using a 50% banded herbicide application, some forget and mistakenly apply twice the recommended rate.

The importance of intraspecific competition control of wild natural pine is often overlooked since it can reduce the long-term survival of genetically improved trees. The stocking can be so intense (Appendix A Table A1) as to require a precommercial thinning at age two to three. Without an early thinning, the superior genetics is often diluted as it is nearly impossible to distinguish between wild pines and improved genotypes when trees are five-years-old.

### 7.1.1. Mowing

When performed correctly, mowing nine times during the first three years after planting can reduce weed biomass without killing planted pine seedlings [134]. However, in some locations, mowing can reduce seedling survival [135].

### 7.1.2. Herbicides

After planting, about 85% of the area is treated with herbicides in the SUS [8]. Appropriate use of herbicides helps to explain the increase in survival since 1955 (Figure 1). Although errors in application can result in lower survival, the frequency of these mistakes declines when using experienced applicators.

Poor communication can result in herbicide-related seedling mortality. In one case, a contractor was told to treat 10 ha with a 50% band of hexazinone at a rate of 1.1 kg ha$^{-1}$. The contractor was accustomed to applying a 2-m band of fertilizers (tree rows spaced 4 m apart), and this "fertilizer method" was also used when applying herbicides. Therefore, instead of applying 5.5 kg of hexazinone to 10 gross ha, the contractor purchased 11 kg and applied it in bands to 10 gross ha. This mistake applied twice the amount of herbicide as intended. When herbicides are applied in bands, it is important to clarify how much herbicide to apply to the treated area (instead of the total area). Accurate records are important when herbicide errors kill seedlings.

Seedling mortality may occur due to backpack application errors when herbicide products are mixed in the individual sprayer but not properly agitated to create a uniform tank mixture. Poor agitation may cause some herbicides to settle in the bottom of the

sprayer, which applies a potentially toxic rate initially followed by an inadequate rate near the end of the application.

Root-inhibiting herbicides might increase pine mortality when rainfall after planting is below normal. Some organizations apply a tank mix of herbaceous and woody herbicides in October and then plant pine seedlings in November. When the herbaceous herbicide does not inhibit root growth, the reduction in competition can increase survival. However, when rainfall is limited after planting, mortality of newly planted seedlings increases when the root inhibitor reduces the uptake of soil moisture.

### 7.1.3. Applying a Mulch

Mulch can help reduce loss of soil moisture and, in some years, a mulch will increase survival. In one dry year, adding mulch around pine seedlings increased survival by as much as 26% [136,137]. When rain is near normal, survival in mulched areas increases by 5% [134].

### 7.1.4. Prescribed Fire

Prescribed burns can reduce fuel loads and can lower the risk of mortality from wildfires. However, if conducted too soon after planting, prescribed fires can reduce the survival of planted pine seedlings [138,139].

### *7.2. Browse Protection*

Browsing can kill pine seedlings when they are pulled out of the planting hole or when the entire shoot is decapitated. Once seedlings become established, however, browsing 50% of the new shoots may not reduce survival. Browsing pines by rabbits can increase mortality [140], but sometime browsing soon after planting will reduce the seedling height and increase survival [141]. When browsing removes the shoot (height after browsing 6 cm), death could occur when weeds overtop pines with suppressed height and root growth. However, at moisture-deficient sites, browsing (height after browsing 16 cm) reduces the rate of transpiration, and this can increase survival [142].

In some areas, deer will browse more on container-grown stock than on bareroot stock [143]. Gopher herbivory of planted seedlings is common in the ponderosa pine range [144].

Cows can kill 10% of pine seedlings by trampling, dislodging and eating pine seedlings [145,146]. At one site that was bedded, cows walked in the furrows and ate all pines that were planted in the furrow but did not injure or dislodge seedlings that were planted on the top of the beds (personal communication John Mexal). Afforestation of pastures following long-term grazing of livestock can result in significant mortality during droughts due to soil compaction, which limits planting dept and root growth.

Carbohydrates in the roots of longleaf pine often attract hogs, which pull up seedlings after they are planted. "Hogs probably have ruined more longleaf plantations than drought, pocket gophers, leaf-cutting ants, and brown spot combined" [110]. In Louisiana, hogs caused the complete destruction of 364 ha of slash pine and recently damaged loblolly pine at two sites in Texas [67].

### 7.2.1. Tree Tubes

Tree tubes are used to reduce damage from browsing. For eastern white pine, installing plastic tree tubes might increase survival by 11% [147] or decreases it by 27% [148]. The environment in the tube likely explains why mortality can increase when using unvented tubes. The daytime temperature inside the tube is higher than the ambient, and this might explain why survival is low at some locations. Use of tubes can cost more than $3.00 per plant [149] and, therefore, use is limited to governments, universities, wealthy landowners or small areas.

### 7.2.2. Fences

Deer populations have increased in several areas since 1950. Hunting clubs use fences to keep deer in, and foresters sometimes use fences to keep deer out. In the northeastern region of the USA, fences are commonly used to protect natural regeneration from deer browsing of high-value hardwoods.

### 7.2.3. Repellents

There are various types of chemical repellents on the market, but most wear out over time. Visual repellents include bud caps, which are fabricated using paper, mesh or cloth [149,150].

### 7.3. Insect Control

Planting insecticide-treated seedlings adheres to the precautionary principle, and some landowners purchase insecticide-treated seedlings. It might cost $5 to treat 1000 pine seedlings before planting. In contrast, sometimes insecticides are applied after the landowner notices mortality. Insects that kill pine seedlings soon after planting in the SUS include regeneration weevils, grubs and leaf-cutting ants.

### 7.3.1. Debarking Weevils

Weevils cause pine seedling mortality in Sweden, South Africa, the UK and the SUS [130,131,151,152]. When logging occurs after July in the SUS, planting is usually delayed for 9 months to reduce the risk of mortality from Pales weevil (*Hylobius pales*). When logging occurs in September or later, and seedlings are planted soon after, injury to seedlings can exceed 50% [151,153]. However, even with this practice, seedling mortality due to reproduction weevils can be significant (Figure 20), especially when non-merchantable (generally < 13 cm DBH) pine residuals are present after logging. When non-merchantable pines remain after harvesting, chemical site preparation and burning after July can attract weevils, which can increase the mortality of newly planted seedlings.

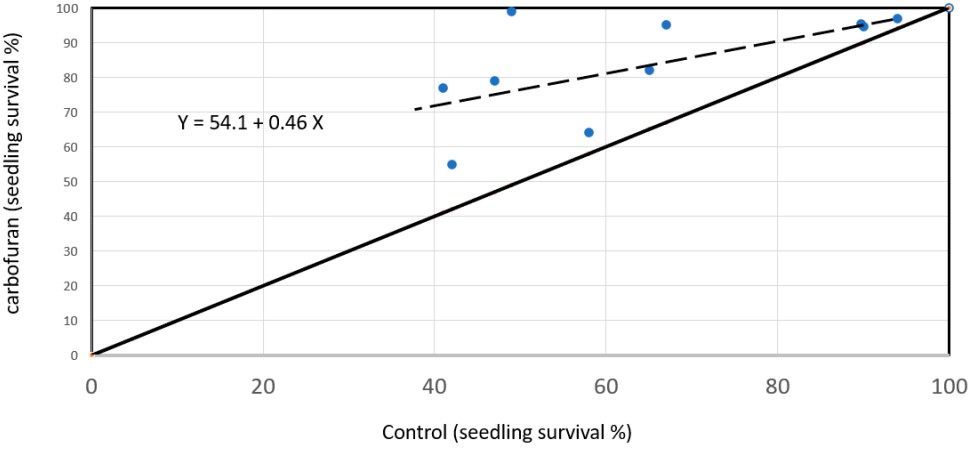

**Figure 20.** Treating seedlings with an effective soil-applied insecticide at the time of planting increased the survival of bareroot loblolly pine and slash pine. Data from [27,151,154]. Although carbofuran is no longer tested or sold in the USA, a rate of 1.8 g per seedling increased seedling survival. Carbofuran is banned in Canada and the EU, but it currently is used to control weevils in row crops in several countries. Points above the solid line represent sites where carbofuran increased survival. $p > |t| = 0.004$ for intercept and 0.052 for slope; $R^2 = 0.39$.

### 7.3.2. White Grubs

White grubs is a term that includes several insect species, and when populations are high enough, they can kill newly planted pine seedlings [130,155–157]. Scalping soil in old fields before planting is one method that can improve survival by moving grubs away

from planting holes. The chance of mortality may be greater when seedlings are planted on abandoned farmland [27,158], and the reason is partly due to grubs feeding on roots.

### 7.3.3. Leaf-Cutting Ants

Leaf-cutting ants in the *Atta* and *Acromyrmex* genera have killed pine seedlings in North and South America. In 2016, Texas leafcutter ants (*Atta texana*) were the leading cause of mortality for pines planted on a site in Cherokee County, TX. Mortality ranged from 80% to 99%. "Leafcutter ant damage was observed as early as 1 month after planting and continued through the third year" [67].

### 7.3.4. Tip-Moths

Insecticides are routinely applied to pine seedlings in order to increase height growth on sites known to contain the Nantucket pine tip moth (*Rhyacionia frustrana* Scudder). On some sites, treatment with an insecticide [159] can increase seedling survival (Figure 21).

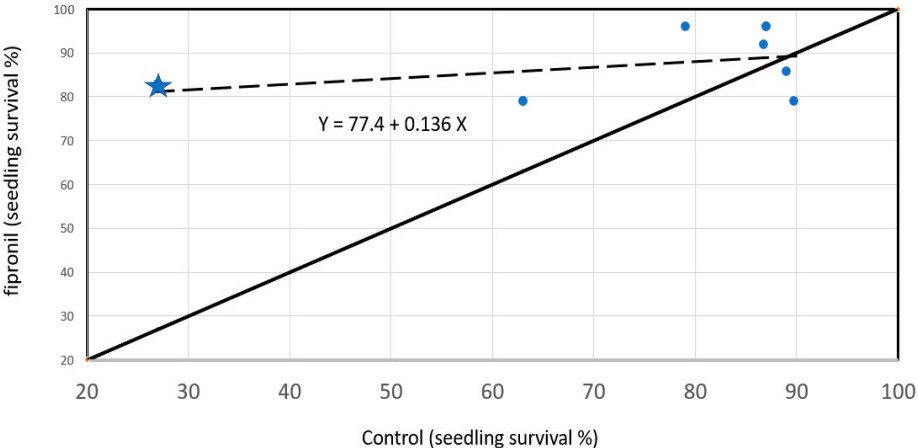

**Figure 21.** A dilution of fipronil was applied to the root zone within one week of planting bareroot loblolly pine seedlings (March 2008) at seven sites in Virginia [153]. The insecticide increased survival on three sites ($\alpha = 0.10$) and reduced foliar damage to seedlings at six sites. Harvest of pines prior to planting was completed before July 2007 except for one site harvested in October 2007 (blue star). Points above the solid line represent sites where fipronil increased survival. $p > |t| = 0.0007$ for intercept and 0.42 for slope; $R^2 = 0.13$.

### 7.3.5. Other Insects

Various other insects also feed on pines. Sometimes the redheaded pine sawfly (*Neodiprion lecontei* [Fitch]) can cause 35% defoliation [160], and too much defoliation can kill small seedlings [157].

### *7.4. Nematodes*

Some nematodes are benign, while others can kill pine seedlings in nurseries and in pine stands. Inoculations with the pine wood nematode killed two-year-old pine seedlings in a bareroot nursery [161]. It is possible that when insecticide treatments increased seedling survival on sites without pales weevil, the increase is due to additional root growth due to a reduction in nematodes. At one site in Florida, the endoparasitic pine cystoid nematode (*Meloidodera floridensis* Chitwood, Hannon and Esser) occurred at high levels and treatment with nematicides increased survival by 8 to 10% [162].

### *7.5. Fungi*

At some sites, a lack of ectomycorrhizal fungi can reduce the chance of seedling survival, but healthy pine seedlings have sufficient mycorrhiza when they pass by the nursery gate. At some locations, pathogens such as *Fusarium circinatum* and *Scirrhia*

*acicula* can kill newly planted seedlings [130], and some storage fungi can quickly lower a seedling's root-growth potential.

## 8. Weather after Planting

Adverse weather can kill pines within a year of transplanting. Dry soil, wet, anaerobic soil, frost heaving, de-acclimation cycles and hail can kill recently planted seedlings. When soil is saturated from rain, strong winds will sometimes topple seedlings during the first five years after planting.

### 8.1. Rain

Regions with low rainfall typically have few pine nurseries. For example, Alabama and Arizona average 1400 and 320 mm/year, respectively. As a result, Alabama grew 116 million conifer seedlings in 2020 compared to less than a thousand seedlings in Arizona. Average rainfall determines economic returns from planting pines. In dry regions in the West, the US government typically supplies most of the funds to grow and plant pines.

#### 8.1.1. Insufficient Rain

Low seedling survival is expected when limited rain occurs during hot summer months. For example, seedling survival averaged 87% in Mississippi and 25% in Louisiana when rainfall (May to September 1980) averaged 122 and 25 mm/month, respectively (Figure 22).

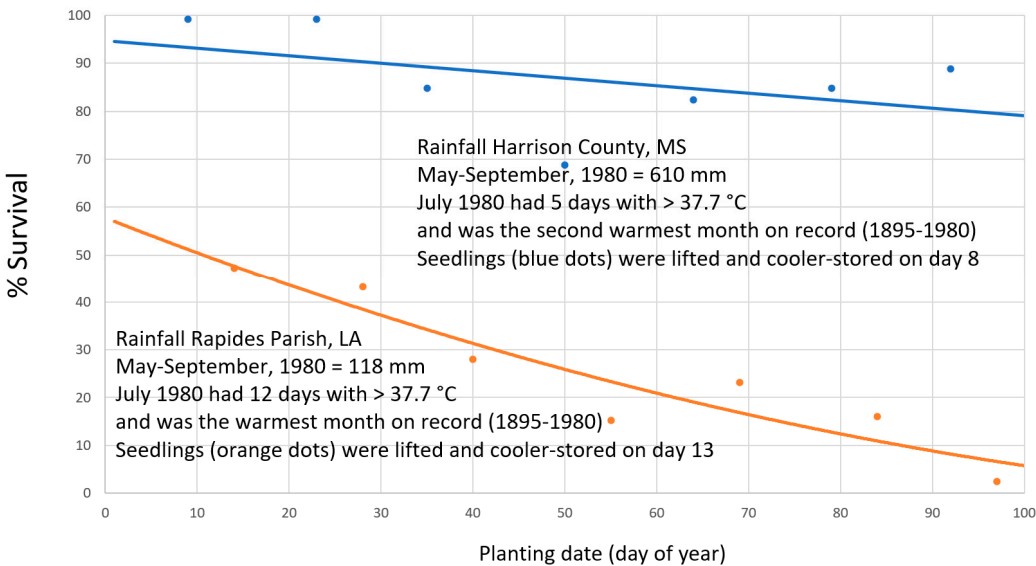

**Figure 22.** Bareroot longleaf pine seedlings were treated with clay slurry just before hand-planting on sites at the Palustris Experimental Station (Louisiana) and the Harrison Experimental Station (Mississippi). Data from [163] Seedling survival was recorded in October of 1981. For Harrison County, $p > |t| = 0.17$ for intercept and 0.31 for slope; $R^2 = 0.20$. For Rapides Parish, $p > |t| = 0.0001$ for intercept and 0.002 for slope; $R^2 = 0.87$.

In some regions (Figure 23), seedling survival for pine is positively related to the Palmer-Drought-Severity-Index (PDSI). The PDSI uses readily available temperature and precipitation data to estimate relative soil dryness. The PDSI typically has a range of −4 (dry) to +4 (wet), but more extreme values are possible. Current and predicted PDSI values for planting chances are available for use in determining if tree planting should begin or cease (https://www.drought.gov/states/texas) (accessed on 1 December 2022). PDSI at the time of planting is likely a better predictor of pine survival [164] than air temperature recorded at the time seedlings are placed in a planting hole. In East Texas, a three-unit increase in PDSI might increase survival by 10% [164].

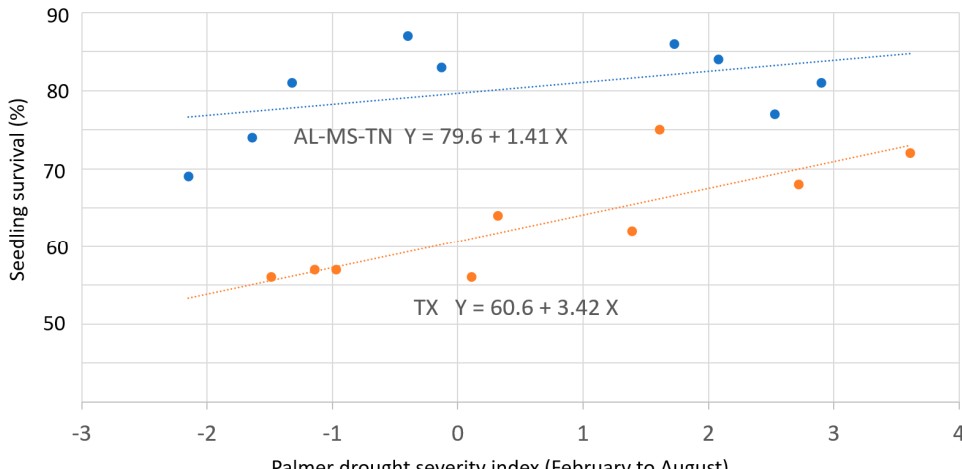

**Figure 23.** Survival of loblolly pine seedlings is related to the Palmer-Drought-Severity-Index (PDSI) calculated for the period of February to August. Data from [164]. Data from Alabama, Mississippi and Tennessee included company survival data from 690 locations and data from East Texas were obtained from a survey of 3570 sites. All seedlings were planted from 1983 to 1992 (one average per year). In East Texas, the year with the greatest survival (1989) had the highest rainfall (February to August—NOAA climate division 4; 130 mm/month; PDSI = 1.61). The year with the lowest rainfall in East Texas (1988) averaged 68 mm/month (PDSI = −1.49). For East Texas, $p > |t| = 0.0001$ for intercept and 0.0047 for slope; $R^2 = 0.70$. For AL-TN-MS, $p > |t| = 0.0001$ for intercept and 0.21 for slope; $R^2 = 0.21$.

When rainfall is inadequate, shade can slow evapotranspiration and increase survival. Therefore, an interaction exists between soil moisture and benefits from shade. In dry years, shade preserves soil moisture, reduces evapotranspiration and may increase first-year survival. In years with above-average rainfall, shade will reduce photosynthesis and can the reduce survival of newly planted seedlings.

When rainfall is adequate, over 90% of shade-tolerant Douglas-fir seedlings survive transplanting without any shade. In contrast, when rainfall is below normal, shade can slow the loss of soil moisture, and this can increase seedling survival (Figure 24). Since artificial shade is expensive (perhaps $0.40 per screen) and accurately predicting rainfall amounts is difficult, tree planters in western states may select naturally shady planting spots [21,98].

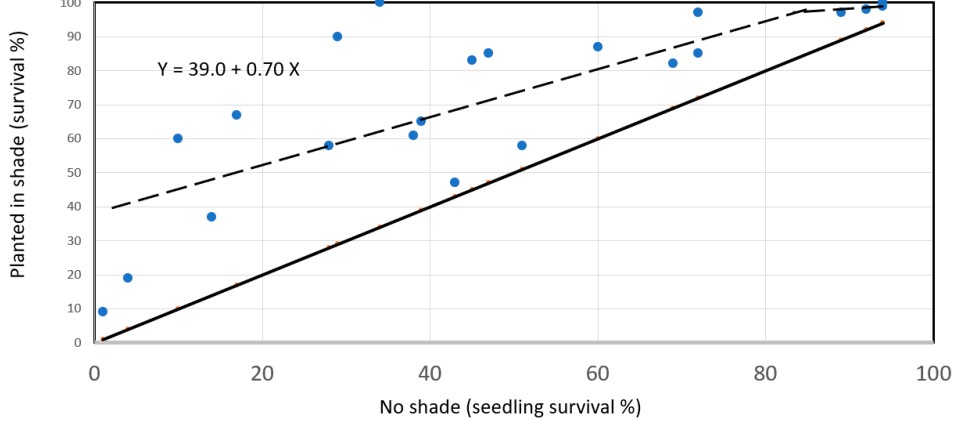

**Figure 24.** At reforestation sites in Oregon, shade can increase the survival of shade-tolerant Douglas-fir seedlings. Data from [165–168]. The dashed line represents the regression equation ($n = 22$) and the solid line represents no benefit from shade. Points above the solid line represent sites where shade increased survival. $p > |t| = 0.0001$ for intercept and 0.0001 for slope; $R^2 = 0.61$.

In most rainy years, shade-intolerant pines are usually resistant to solar radiation and so artificial shading devices are rarely necessary after planting [169]. East of the Mississippi river, survival of unshaded, bareroot loblolly pines is typically greater than 80%. In years with adequate rainfall, shade can decrease the survival and growth of container-grown longleaf pine [170]. However, in hot and droughty years, partial shade can reduce transpiration and increase seedling survival one year after planting [110,126,143,170–172]. In a Texas study with limited rainfall, shade increased the survival of loblolly pine by 25% in areas with weeds and 30% in areas treated with two herbicide applications [126].

8.1.2. Too Much Rain

Observations over the past four decades indicate that waterlogged conditions during the year of planting lower the survival of pine seedlings, especially soon after planting. Waterlogged soils can develop when frequent rains occur over an extended period. On well-drained soils, above-average rain might not reduce soil oxygen or seedling survival. However, on fine-textured soils, water may accumulate due to low infiltration (Figure 25). For the southeast region, record amounts of rainfall (>430 mm) fell during the last three months of 2009, 2015 and 2019.

Mortality results on some soils when rainfall exceeded 50 mm/week for a period of three weeks or more during the autumn or spring. Anaerobic conditions can occur quickly when warm soils remain saturated for just a few days. Root growth is slowed, and the transpiration of seedlings is reduced.

When anaerobic soil conditions develop, lenticles are produced on the stem of pines near the root-collar or slightly above the soil surface. Although lenticles are not harmful, they can be used to diagnose the cause of low seedling survival (Figure 26). When a regeneration survey is conducted, recording the frequency of lenticles on 100 sampled seedlings may provide a clue as to the cause of unexpected mortality.

In some places, inundation causes complete mortality of loblolly pine [173,174]. In contrast, a mortality of 90% occurred when potted seedlings were flooded for 5 days with salt water [175].

*8.2. Temperature*

Most SUS foresters plant pines in clear-cut areas or on abandoned agricultural fields where soil temperatures fluctuate. Both low and high temperatures can kill newly planted pine seedlings.

8.2.1. Temperatures above 0 °C

Once pine seedlings leave the nursery, they should be stored in cool environments. Some landowners in Virginia might store seedling bundles in unheated buildings for several weeks in January before planting. Alternatively, seedlings are packed into KP bags or in wax-impregnated cardboard boxes. When stored in an unheated warehouse in Mississippi, the temperature in bags in April can exceed 29 °C (Figure 27). When small pine seedlings (11.6 cm tall) are stored in a cardboard box at 15 °C in Finland, mortality increases after 2 weeks of storage [96].

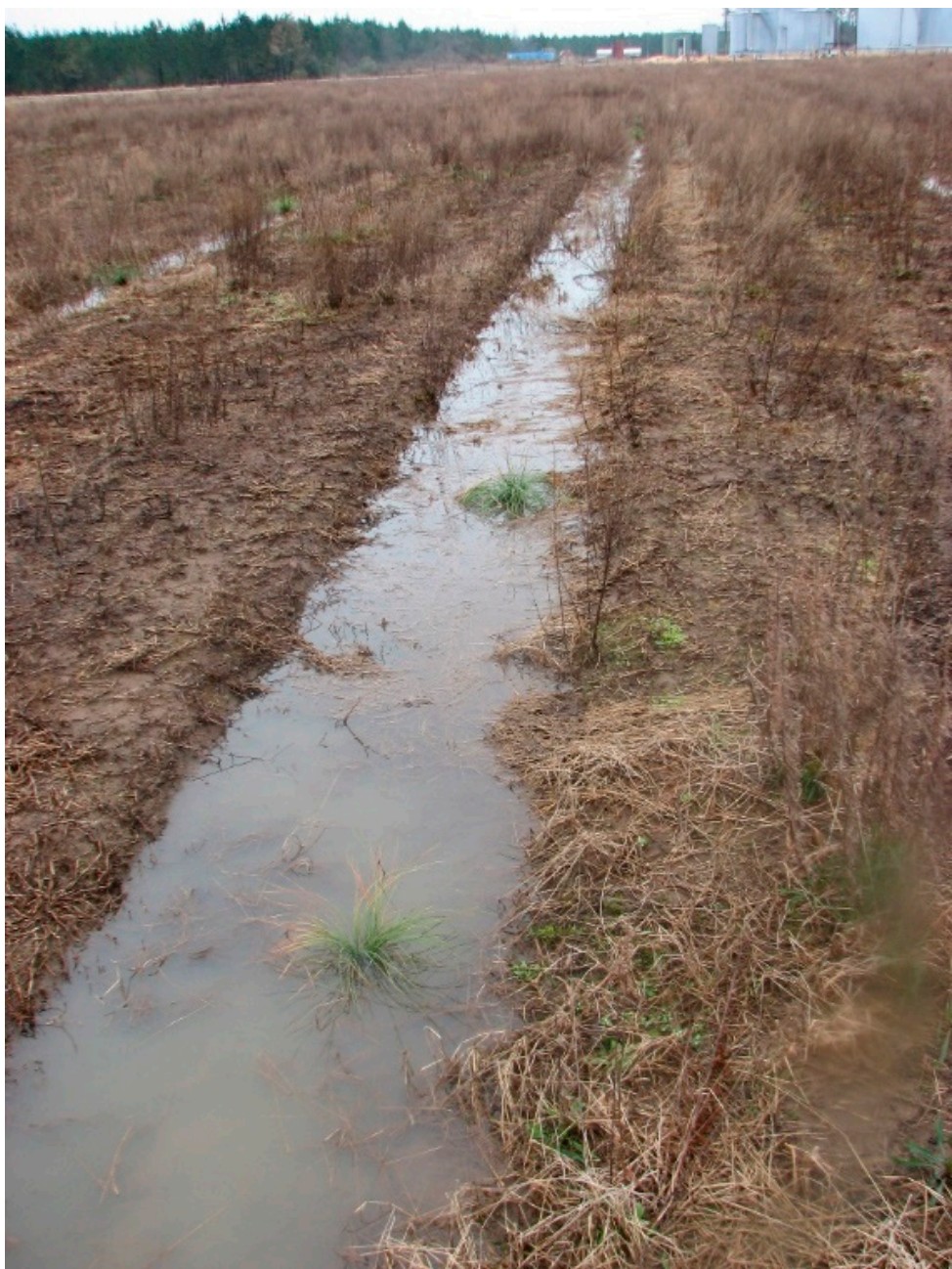

**Figure 25.** Scalping a fine-textured soil resulted in standing water which killed longleaf pine seedlings. Data from [68]. Seedlings were machine planted on 29 November 2007, and in April 2008, 243 mm of rain occurred over a 15-day period. A year later, March rain occurred on the 4th (41.1 mm), 7th (81.5 mm) and 12th (4.3 mm) and the photo was taken 13 March 2008. Scalping increased the mortality of bareroot stock (73% and 91% survival on scalped and non-scalped areas, respectively).

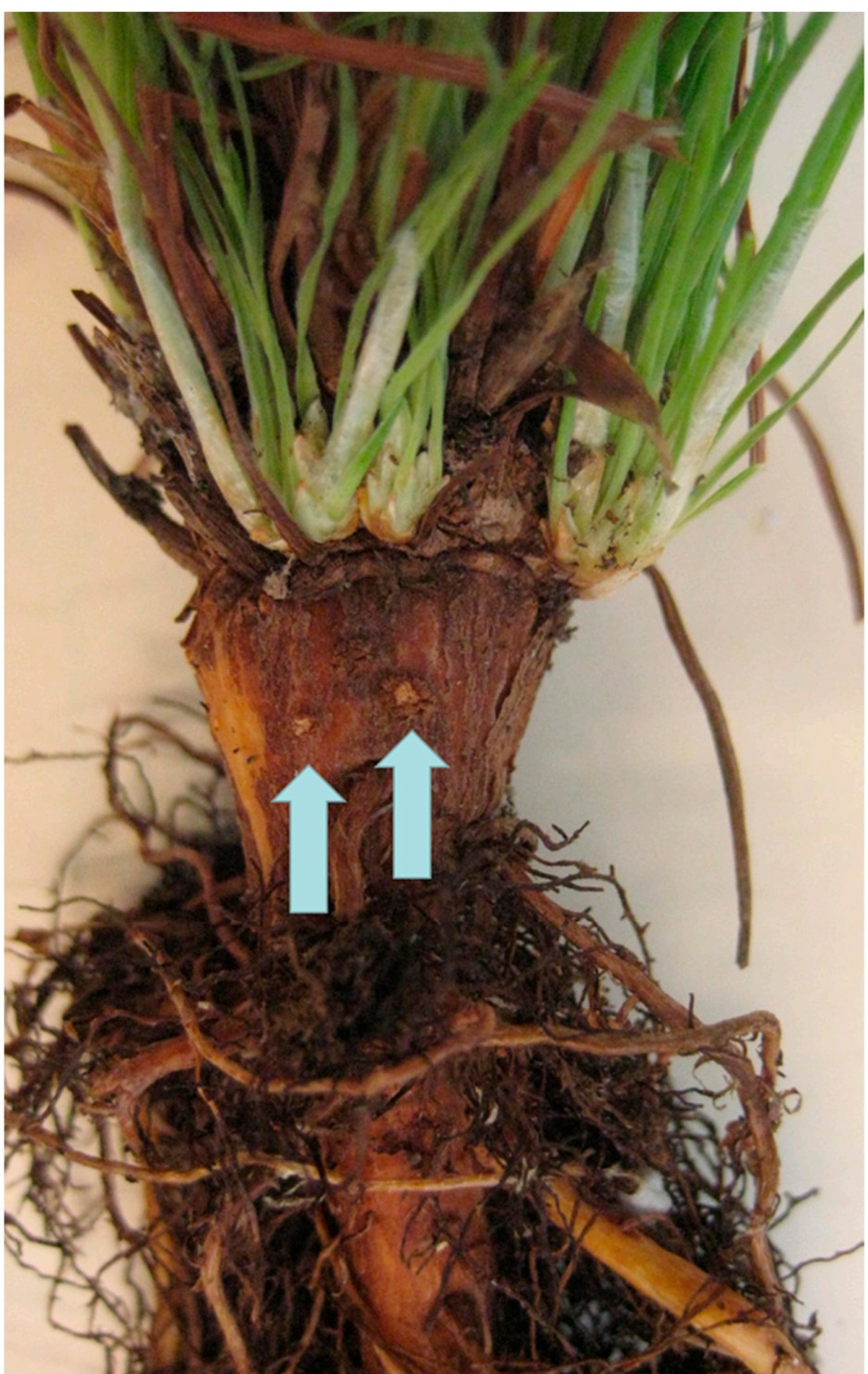

**Figure 26.** Example of two lenticles on the root of a container-grown longleaf pine seedling.

Planting conditions may be marginal for pine seedlings when the air temperature during outplanting is 24 to 29 °C, and relative humidity is 30% to 50% [38,121]. In one trial, mortality increased by more than 18% when the air temperature at planting exceeded 24 °C (Figure 28). Temperatures in bags can be 10 °C higher than air temperatures when bags are stored for 4 to 6 h outside in the direct sun. To reduce the chance of heating seedlings in bags or drying roots before planting, a few foresters suggest tree planting should be stopped when afternoon temperatures reach 30 °C. Many seedlings likely die due to heat buildup when bags are not kept cool.

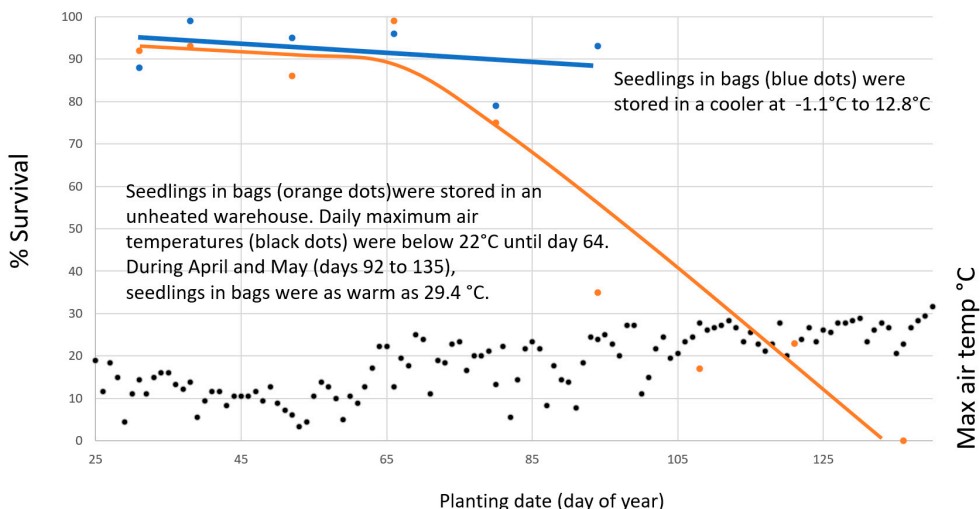

**Figure 27.** Survival of loblolly pine seedlings is related to the storage method. Data from [176]. Bareroot seedlings lifted on 9 January and cooler-stored for 12 weeks in kraft-polyethylene bags (containing 454 g of sphagnum moss) exhibited good survival when planted at Oxford, Mississippi, on 3 April 1964. The temperature of seedlings stored in an unheated building exceeded 25 °C, and they did not survive well when stored for more than 9 weeks [176]. For the survival of seedlings stored in bags, $p > |t| = 0.0001$ for intercept and $0.0003$ for slope; $R^2 = 0.87$.

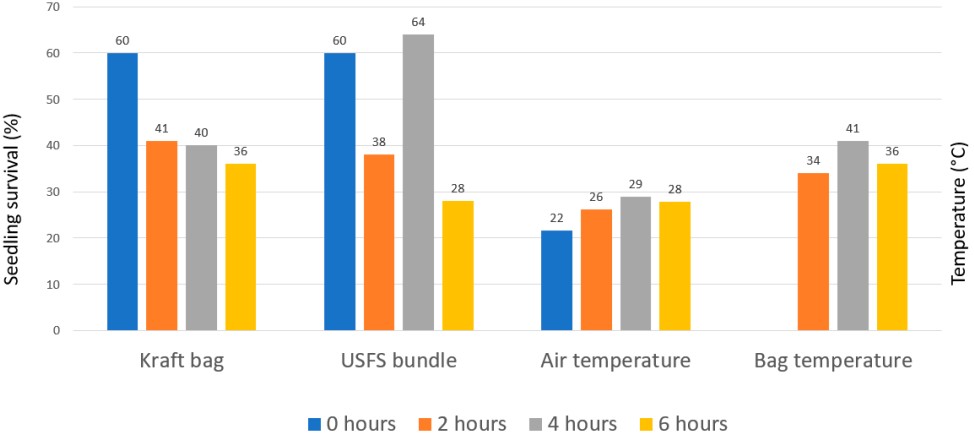

**Figure 28.** Survival of loblolly pine seedlings lifted on 13 March, stored in a cooler for 47 days at 3.3 °C in either kraft-polyethylene bags or bundles, and planted on 29 April 1970. Data from [177]. Each package contained 1000 bareroot seedlings, and seedlings were extracted and planted from packages at 9:30 a.m. (zero exposure), 11:30 a.m. (2 h), 1:30 p.m. (4 h) and 3:30 p.m. (6 h of exposure while in the bag or bundle). The interaction between the packing method and the exposure time was not significant ($\alpha = 0.05$). The least significant difference ($\alpha = 0.05$) for a one-tail test is −17% and ±32% for a two-tailed test.

The recommendation to stop planting at 30 °C relates to drying exposed roots prior to planting and not to lethal air temperatures after planting stock into moist soil. Five months after planting in February [136], loblolly pine seedlings tolerated a brief air temperature of 67 °C (2.5 cm above a mulch). For some northern pines in a laboratory, mortality occurred when air temperatures exceeded 48 °C for 5 h [178].

Most tree planters in the SUS prefer planting seedlings before May, but sometimes pines are planted during the summer [104,144,179–181]. Planting seedlings in hot and dry soil in July in drought conditions D2 to D4 can kill container-grown pine seedlings [182,183]. When seedlings are handled with care, pines can be planted in moist soil during the summer. For example, at an airport in Pensacola, FL, the month of July 1966 had 80 mm of rain, and each day temperatures in the afternoon exceeded 29 °C (Figure 29). Although the

temperature reached 36 °C on July 8th, the survival of bareroot slash pines planted in that month averaged 80%. During 1966, 1.3 million bareroot slash pine seedlings (seed sown in a nursery in mid-November 1965) were planted in the summer in Florida and South Georgia [181].

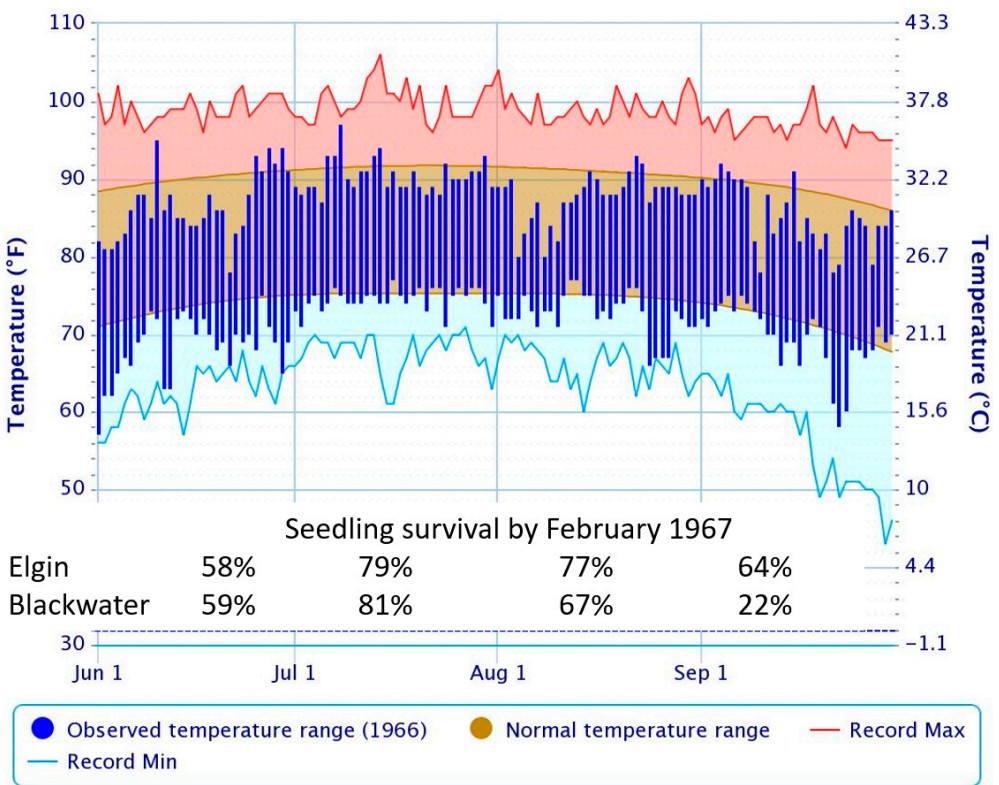

**Figure 29.** Seedling survivals for bareroot slash pine in Florida vary by planting location and month. Data from [181]. Survival of June planted stock was 58% when planted at the Elgin Air Force Base and 59% when planted at the Blackwater State Forest in Florida. Air temperatures for the Pensacola regional airport for the year 1966 were provided courtesy of the Applied Climate Information System. The highest temperature of the year was 36.1 °C on 8 July 1966. Data for maximum and minimum temperatures began in 1948.

Many pine species will tolerate transitory air temperatures of 45 °C during summer months, and some can tolerate 56 °C for a brief period [184,185]. During a prescribed burn, longleaf pine seedlings can survive brief air temperatures of 400 °C [50]. When moisture is adequate, pine roots can tolerate 45 °C [178].

8.2.2. Temperatures below 0 °C

When soil moisture is adequate, recently planted loblolly or longleaf pines are more likely killed by a sudden December freeze than by air temperatures during planting of 30 to 35 °C. Temperatures below −8 °C have killed seedlings in the SUS from October to March. One week of freezer storage at −6.7 °C can reduce seedling survival by 44% [186], and two weeks of freezer storage at −3 °C killed all the clones of loblolly pine [187]. Likewise, in a growth-room trial, more Douglas-fir seedlings were killed after 2 h of −8 °C temperature than by 24 h of growth-chamber storage at 32 °C [188].

We classify freeze events into three types; preacclimation, acclimation and de-acclimation [189]. In the United States, a preacclimation freeze might occur in October or November when seedlings have not received a sufficient level of freeze tolerance. Acclimated freezes occur after the winter solstice and before warm weather in February (Appendix A Table A1). A de-acclimation freeze occurs when enough warm nighttime temperatures decrease the level of freeze tolerance. The frequency of de-acclimation

freezes has likely increased since 1950. In a region stretching from Mississippi to North Carolina, the last hard freeze tends to occur later now when compared to the first part of the 20th century [190].

Unimproved loblolly pine seedlings with sufficient chilling usually tolerate −8 °C freeze events if they have not been de-acclimated due to warm nighttime temperatures (Figure 30).

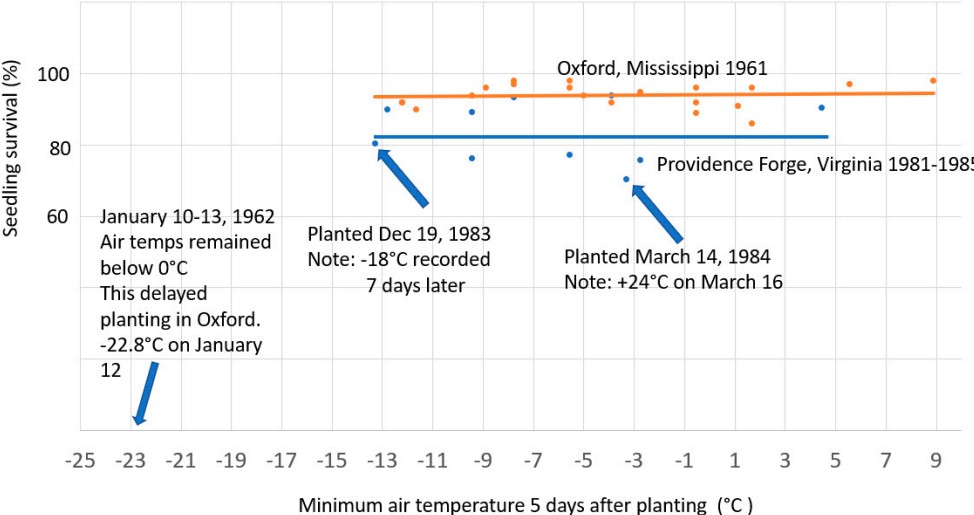

**Figure 30.** Acclimation freezes above −14 °C had minimal effect on the survival of bareroot loblolly pine seedlings planted from 16 December 1981 to 3 March 1985 (blue dots). Virginia data are from [191], and Mississippi data (orange dots) are from [192]. The cambial injury was not examined, and shoot growth may have been reduced. The *X*-axis represents the minimum air temperatures during the first 120 h after planting. Air temperatures above −14 °C and below freezing apparently did not reduce survival since the seedlings had sufficient chilling. In contrast, when the soil remains frozen for several days, seedling mortality can increase due to loss of plant water [110]. Blue and orange dots represent first- and third-year survival, respectively. Temperature data from Williamsburg, Virginia and University, Mississippi. Seedlings in Mississippi had little storage, while those planted in March in Virginia were stored in a cooler for 2 to 3 months. The air temperature reached −22.8 °C on the morning of 12 January 1962.

The preferred time to plant pine seedlings in Nebraska is in the spring when the risk of freezing weather is low. In contrast, in Alabama, the preferred time to plant container-grown longleaf pine is mid-September to late November, when −8 °C freezes are rare [100]. In contrast, the survival of longleaf seedling planted in December may average less than 70% due to freezing roots prior to shipping [193] or due to hard freezes just after outplanting (Table 1).

When called to investigate the reason why genetically improved pine seedlings died, we examine the cambium on roots and stems and dead needles and buds. When a de-acclimation freeze injures the cambium, the xylem continues to transport water at a slow rate, and pine needles remain green until warm temperatures arrive. New root growth is reduced, and when evaporation exceeds transpiration, seedlings die. With a de-acclimation freeze event, the landowner might not be convinced a −5 °C freeze (reported on the radio) was the problem. Some landowners were looking to blame planting quality, seedling handling or a nursery cultural practice. In some cases, landowners believed mortality in a frost pocket was not from a −7 °C freeze since they historically achieved good survival with acclimated seedlings that tolerated temperatures below −8 °C (Figure 12).

In Finland, sometimes half of the 155 million container-grown seedlings remain outdoors during winter [194], and a similar amount remains outdoors in winter in the SUS. However, container-grown pine seedlings stored outside can be injured due to freezes [193,195,196]. In Virginia, temperatures also can reach −22 °C in winter and,

therefore, seedlings are extracted from containers, placed in boxes in October, and stored in coolers until sold. Some customers purchase seedlings and then plant just prior to freeze events. Others purchase seedlings and then plant in April after seedlings have been stored for 10 weeks in the cooler. In 2019, the survival of seedlings planted just before a freeze ranged from 43% to 67%, while those planted in April had 94% survival [112].

Sometimes landowners store seedlings in unheated buildings before planting. When temperatures remain below $-1\,°C$ for a three-day period, seedlings in bags or bundles can freeze solid. Seedlings may survive if allowed to completely thaw before moving (Figure 31).

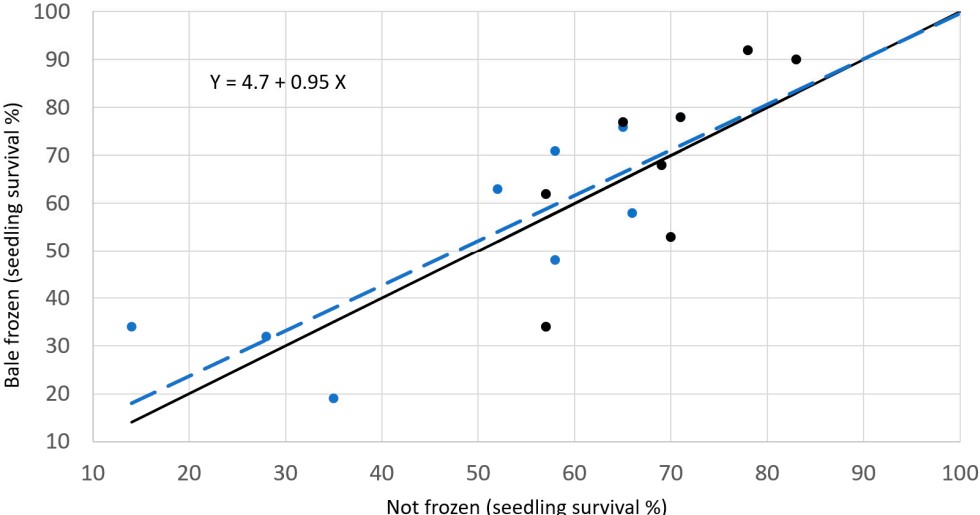

**Figure 31.** Seedling survivals for genetically unimproved bareroot loblolly pine packed into bundles (2000 seedlings per bail) in January of 1970 at the New Kent nursery in Virginia. Data from [197]. Bales were naturally frozen ($-11$ to $-12\,°C$) outdoors on January 19 and 20 (blue dots), or February 1 to 3 (black dots). Frozen seedlings were thawed in an unheated building (beginning 23 January or 8 February) and outplanted 34 to 48 days after lifting. When planted in February, survival averaged 69%, and freezing made no difference to survival. When planted in January, non-frozen and frozen seedlings averaged 47% and 50%, respectively (LSD$_{05}$ = 10.8%; two-tailed test). Points >8% below the solid line represent cases where freezing reduced survival. $p > |t|$ = 0.68 for intercept and 0.002 for slope; R$^2$ = 0.65.

Even after a $-13\,°C$ freeze, 80% of sensitive pine genotypes died after roots in the containers froze (Figure 32). For this reason, planting loblolly pine genotypes too far north of their seed origin (more than $-5.5\,°C$ in average winter temperature minimum) is not recommended [198].

Some people believe predictions about weather events in 2050 are equivalent to facts [199–201]. In fact, some advocate planting genotypes more than 100 km north of their natural range [202,203]. In fact, more than 29,000 ha of loblolly pine seedlings have been planted north of their natural range. In one trial, loblolly pine planted 700 km north of the origin had 9% survival 37 years after planting [46]. At that age, survival near 50% would be expected for a non-thinned stand planted in Mississippi.

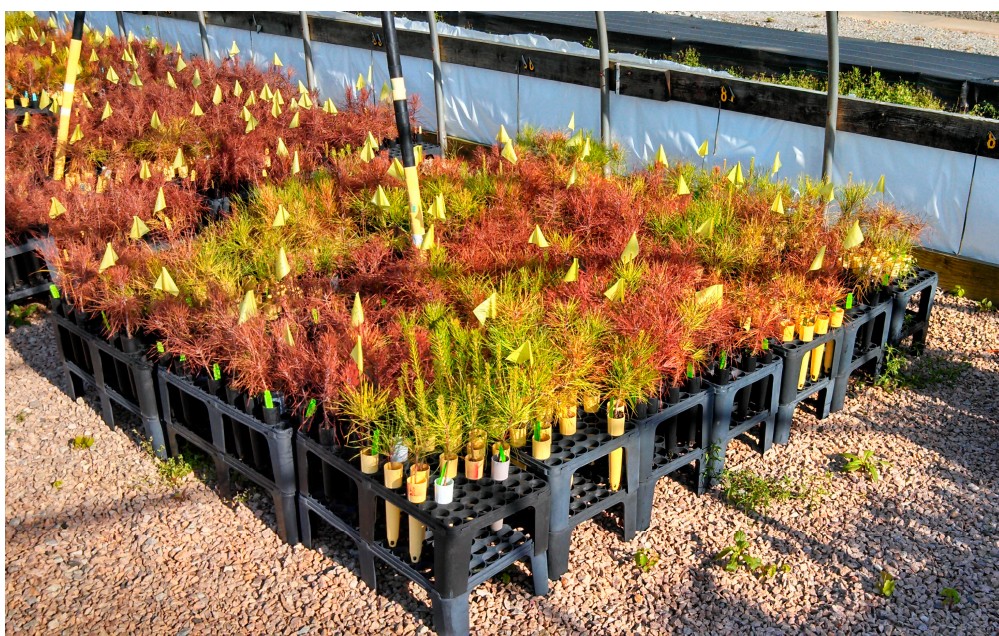

**Figure 32.** After the wind destroyed a protective plastic cover, rooted cuttings of loblolly pine were killed by a 30 January 2014 freeze (−13 °C). Although average survival was 30%, one northern genotype had 80% survival [204]. Reproduced with permission from Steve McKeand [204].

### 8.2.3. Winter Desiccation

Winter desiccation can occur when the soil remains frozen for several days and pine seedlings are not protected by a cover of snow. When transpiration is reduced due to frozen soil, a combination of wind and sun can desiccate seedlings [205]. This type of injury is rare in the SUS, but four days of below-freezing weather (16–19 January 1977) desiccated exposed seedlings at a nursery in Virginia [206].

### 8.2.4. Frost Heaving

Frost heaving in plantations can kill bareroot and container-grown pine seedlings [207,208]. One way to reduce mortality caused by frost heaving is to make a deeper hole and place the root collar 10 to 15 cm below ground and cover the roots with more soil [122,209]. Planting the root-collar of pine only 5 cm below the ground will likely have a minimal effect on frost heave [210]. Even so, in one trial, planting small container seedlings (11 cm taproot) 1.5 cm deeper increased survival by 20% [211] and frost heaving (5 to 12%) did not occur when the root-collar of pine seedlings was planted 11 cm below the soil surface [212]. Frost heaving may be less of a problem with bareroot seedlings with 18-cm taproots [110] because they require deeper holes than seedlings with 11-cm taproots.

### 8.3. Wind

Naturally regenerated pines typically withstand high winds due to a strong taproot. In contrast, high winds sometimes topple saplings when fast-growing pines are growing in rain-soaked soils [213]. Typically, mortality is near zero when pine lean is only 20° from vertical. However, when pines are prostrate and lying on the ground, mortality increases. For example, hurricane Michael (October 2018) laid 90% of container-grown pines horizontally on the ground [214]. At one site in Brunswick County, North Carolina, the mortality rate exceeded 20%.

### 8.4. Hail

Hail can damage young pines, and sometimes the damage is minor, and mortality is not increased. When hail is the size of eggs, young seedlings can be killed, and those not killed can attract insects [215]. Sometimes hail might account for 2% or more of early

mortality [216,217]. In Australia, the area affected by hailstorms can amount to 300 to 400 ha year$^{-1}$ or about 0.06% of the total plantation area [218].

## 9. Replanting

When initial mortality is greater than 10% or 20%, some pulpwood companies in Africa will plant more seedlings in the blank spots (a practice known as blanking, beating up or interplanting [219]). Often the second planting is conducted within three months of the first planting [10]. This is because blanking one year later seldom adds to the volume of production at harvest, and the added cost will likely not be recovered [220]. Some landowners would rather focus on maximizing initial seedling survival instead of investing additional time and expense on blanking. For loblolly pine, blanking typically has a benefit/cost ratio less than 1. For example, when 50% mortality occurs two weeks after planting, this might result in 741 live seedlings ha$^{-1}$. In this case, blanking might increase stocking to 1482 ha$^{-1}$ (at a blanking cost of $150 ha$^{-1}$ at week 2) without increasing wood volume at age 24 years [221]. In other words, the cost of blanking would lower the land expectation value by $150 with no gain in sawtimber production. Typically, most private landowners in the SUS do not blank since sawlog production is near optimal when seedlings are planted at 741 ha$^{-1}$ [222,223].

## 10. Cost of Improving Survival

Many landowners are willing to spend $100 ha$^{-1}$ to increase seedling growth with either fertilizers or herbicides, but some are reluctant to spend additional money to increase the probability of seedling survival. This is understandable when treatments like mulch or organic amendment costs $2000 ha$^{-1}$ (Appendix A Table A2), but would they be willing to pay $75 ha$^{-1}$ more to plant seedlings deeper with a machine? Machine planting might increase survival by 11% (Figure 10), which might increase initial stocking by 110 trees ha$^{-1}$ (at 1000 ha$^{-1}$). When it costs $ 1 per plant (75 cents per cutting plus 25 cents for planting), then the benefit/cost ratio for increasing stocking by 110 cuttings ha$^{-1}$ would be 1.46 (i.e., $110/$75). However, when it costs $0.20 to plant a seedling (8 cents plus 12 cents to plant), the benefit/cost ratio is less than 0.3 ($22/$75).

Perhaps 6% of pine-plantation landowners in the SUS are willing to pay $345 ha$^{-1}$ to machine plant ($195) container-grown loblolly pine seedlings ($150). In contrast, about 85% are willing to spend $200 ha$^{-1}$ to hand-plant ($125) bareroot stock ($75).

If a low-cost treatment (Appendix A Table A2) could consistently increase survival by a small amount, would landowners plant fewer trees to offset the added cost? A hypothetical comparison is presented in Table 2, where a relatively cheap treatment increases survival by 4% at site A but has no effect on survival at site B. In this example, the extra $10 ha$^{-1}$ cost is offset by planting 50 fewer seedlings ha$^{-1}$. The differences in survival and long-term economics are so small that sampling would likely not be able to detect any significant difference, even when using a one-tailed *t*-test.

**Table 2.** Depending on site, investing an extra $10 ha$^{-1}$ to improve seedling survival of loblolly pine might affect stocking three years after planting. Alternative A increased survival by 4%, while Alternative B did not increase survival. Land expectation value (LEV @6%) estimated from [223].

| Regeneration Method | Seedlings ha$^{-1}$ | Treatment Cost ha$^{-1}$ | Seedling Cost ha$^{-1}$ | Hand Planting Cost ha$^{-1}$ | Total Cost ha$^{-1}$ | Third-Year Survival | Third-Year Stocking ha$^{-1}$ | LEV ha$^{-1}$ |
|---|---|---|---|---|---|---|---|---|
| No treatment | 1300 | 0 | $104 | $156 | $260 | 88% | 1140 | $360 |
| Treatment on site A | 1250 | $10 | $100 | $150 | $260 | 92% | 1150 | $359 |
| Treatment on site B | 1250 | $10 | $100 | $150 | $260 | 88% | 1100 | $364 |

Table 2 suggests that increasing survival by 4% can reduce the land expectation value by 1%. This small difference would not be noticed when stands A and B are harvested

at age 25 years. When too many seedlings are planted, random mortality can increase stand value. This is because, for unthinned plantations, more surviving trees mean more pulpwood, while fewer trees mean more sawlogs. On sites where mortality kills every other planted seedling, stand value can increase by 33% (Figure 33). Many incorrectly assume that increasing survival of planted pines will increase stand value at harvest.

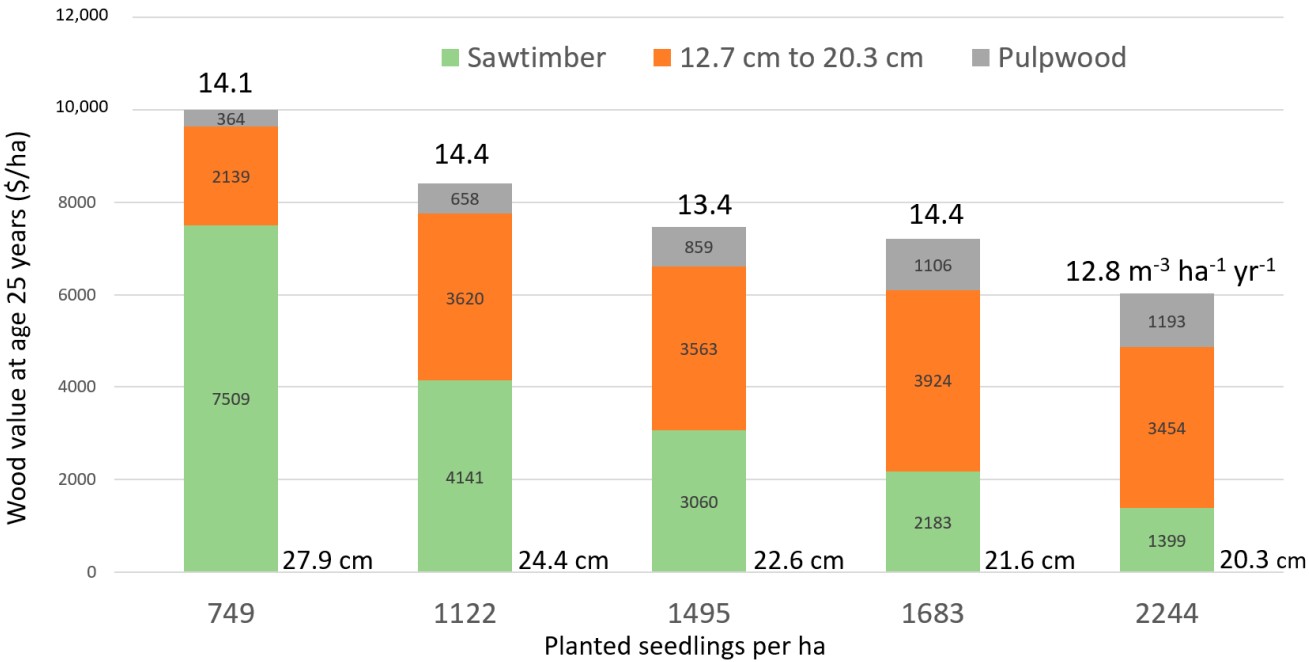

**Figure 33.** A 25-year old, non-thinned loblolly pine stand (749 planted seedlings ha$^{-1}$) has 145% more sawlogs than a stand planted with twice as many seedlings. Data from [222]. Too many surviving seedlings can increase wood production while reducing economic returns. In this example, returns per planted seedling varied from $13.30 to $2.70 ($11 m$^{-3}$ for pulpwood and $35 m$^{-3}$ for sawtimber). The mean annual increment is listed at the top of each bar, and the average DBH is listed at the right of each bar. Half of the seedlings planted at 2244 ha$^{-1}$ died before harvest, while mortality was 22% when seedlings were planted at 749 ha$^{-1}$.

## 11. Recommendations by Researchers

A main goal of regeneration research is to discover treatments that will provide a real increase in seedling survival. Too often, however, researchers use the precautionary principle and do not recommend effective treatments that, on average, increase survival by 5%. Researchers are trained to avoid making a Type I statistical error (i.e., claiming a treatment works when, in reality, it has no effect on survival). Unfortunately, most of us were not trained in how to install a seedling survival test with high statistical power (i.e., LSD ≤ 7% survival). For example, a one-tailed test ($\alpha$ = 0.1) is more powerful than a two-tailed test ($\alpha$ = 0.05), but typically survival means are tested using a low-power, two-tailed test. As a result, many survival trials produce Type II statistical errors (Table 3). When tree planting trials have low statistical power, researchers may consider using more replications [224]. Other methods to increase power include: (1) the use of a one-tailed test [212], (2) the use of a more powerful contrast test (instead of an overall treatment F-test) and (3) conducting survival trials under a roof to reduce soil moisture and increase the onset of mortality [101].

**Table 3.** When trials have low statistical power [224], a real 10% increase in survival is often declared not significant due to the experimental design and/or method of statistical analysis. Increases listed below were not statistically significant ($\alpha$ = 0.05; two-tailed test). LSD = least significant difference.

| Treatment | Survival Increase | LSD $\alpha$ = 0.05 | Reference |
|---|---|---|---|
| | % | % | |
| Subsoiling | 42 | ? | [2] |
| Ripping | 32 | ? | [23] |
| No high burn | 30 | ? | [94] |
| Nursery stock | 22 | 24 | [225] |
| Fertilizer | 21 | 22 | [226] |
| Chilling | 20 | ? | [195] |
| Root gel | 19 | 21 | [87] |
| Planting date | 18 | 19 | [227] |
| Compost | 18 | ? | [228] |
| Herbicide | 15 | 17 | [229] |
| 3-1 plow | 14 | ? | [230] |
| Fungicide | 14 | ? | [25] |
| Root prune | 13 | ? | [53] |
| Planting date | 12 | 16 | [231] |
| Root gel | 11 | 18 | [61] |
| Herbicide | 11 | 22 | [124] |
| Clipping | 11 | 13 | [95] |

Examples of Type II errors in reforestation studies can be found in the literature. In a Mississippi study, an alpha level of 0.01 was used to test for significant soil and weather correlations [192]. After analyzing data, the author wrote, "Data in both years suggested higher mortality for seedlings that encountered freezing weather soon after planting, but again trends did not prove: significant." Due to a small alpha value, the power of the statistical test was low, and the following quote from the paper is an example of a Type II statistical error: "Freezing weather immediately after planting is not lethal."

Seedlings planted on 5 February 1962 experienced a $-7\,^\circ$C freeze on February 7 and survival was 68%. In comparison, survival averaged 78% for six other planting dates in February [192]. The claim that no seedlings were killed by the $-7\,^\circ$C freeze was based on (1) not using a one-tailed *t*-test, (2) using a 0.01 alpha value and (3) not knowing the true reason why each seedling died.

## 12. Discussion

Soon after planting, some landowners want to know why pine mortality was unexpectedly high. Sometimes opinions are provided at no cost without an onsite inspection. When a landowner invites inspectors to quickly examine healthy, dead and dying seedlings, important clues can be documented before they disappear. At several sites, we examined cambium tissue and identified freeze injury as the primary cause of mortality. Weather records can often pinpoint the day of freeze injury. At other sites, lenticels indicated seedlings were exposed to flooded conditions. At several sites, we found piles of roots that indicate new root growth was reduced by trimming roots [69]. At many sites, there were several dead seedlings where the cause of mortality was hard to identify. In Finland, about 24% of mortality is due to unknown factors [232].

Although speculative, we suspect nematodes as a major factor at some old-field sites. Typically, dead and dying roots are not sampled for nematodes, so mortality may be categorized as "unknown." With well-designed studies, future researchers might determine how much of the unknown mortality is due to the feeding on roots by nematodes.

Good seedling survival depends on using practices that are known to not increase mortality. Establishment failure is usually caused by a lack of planning, a lack of adequate supervision, and by not applying simple, inexpensive techniques that are known to increase survival [233].

**Supplementary Materials:** Information on Standardized Precipitation Index (SPI). https://climatedataguide.ucar.edu/climate-data/standardized-precipitation-index-spi (accessed on 17 December 2022). An example of ArborGen's daily seedling planting log is available online at https://tinyurl.com/2cewmc2c (accessed on 17 December 2022).

**Author Contributions:** Conceptualization, D.B.S.; data curation, D.B.S. and A.L.; writing—original draft preparation, D.B.S.; writing—review and editing, D.B.S., T.E.S. and A.L.; graphics, D.B.S.; statistical analyses, D.B.S. All authors have read and agreed to the published version of the manuscript.

**Funding:** This review received no external funding.

**Institutional Review Board Statement:** Not applicable.

**Informed Consent Statement:** Not applicable.

**Data Availability Statement:** Not applicable.

**Acknowledgments:** This manuscript was developed with input from Steve McKeand. Rainfall data were supplied by the National Oceanic and Atmospheric Administration.

**Conflicts of Interest:** The authors declare no conflict of interest.

## Appendix A

**Table A1.** Freeze events, type of freeze and type of injury reported on conifers in the eastern United States.

| Year | Month | Day | °C | Type of Freeze | Type of Injury | Location |
|---|---|---|---|---|---|---|
| 1965 | October | 30 | −8 | Preacclimation | Not available | South Carolina |
| 1991 | November | 5 | −7 | Preacclimation | Needle burn | Alabama |
| 1950 | November | 25 | −22 | Preacclimation | Root cambium | Illinois |
| 1970 | November | 25 | −9 | Preacclimation | Needle burn | Georgia |
| 2006 | December | 9 | −10 | Preacclimation | Root cambium | Alabama |
| 1962 | December | 12 | −27 | Acclimation | Needle burn | Tennessee |
| 1955 | December | 17 | −9 | Preacclimation | Needle burn | South Carolina |
| 2022 | December | 21 | ——————— winter solstice ——————— | | | |
| 1989 | December | 23 | −18 | Acclimation | Needle burn | Alabama |
| 1983 | December | 25 | −12 | Deacclimation | Root cambium | Florida |
| 2004 | January | 7 | −8 | Deacclimation | Root cambium | Georgia |
| 2018 | January | 7 | −25 | Deacclimation | Root cambium | Virginia |
| 2018 | January | 7 | −18 | Deacclimation | Root cambium | North Carolina |
| 1977 | January | 11 | −26 | Acclimation | Needle burn | Kentucky |
| 1962 | January | 12 | −23 | Acclimation | Root cambium | Mississippi |
| 1994 | January | 19 | −14 | Acclimation | Root cambium | Alabama—Mississippi |
| 1957 | January | 19 | −7 | Deacclimation | Needle burn | Florida |

**Table A1.** *Cont.*

| Year | Month | Day | °C | Type of Freeze | Type of Injury | Location |
|------|-------|-----|-----|----------------|----------------|----------|
| 1996 | January | 19 | −9 | Acclimation | Root cambium | Alabama |
| 1985 | January | 21 | −18 | Acclimation | Needle burn | Kentucky—Tennessee |
| 1985 | January | 21 | −37 | Acclimation | Needle burn | North Carolina |
| 2018 | January | 21 | −12 | Acclimation | Root cambium | Virginia |
| 2014 | January | 30 | −13 | Acclimation | Root cambium | North Carolina |
| 1996 | February | 9 | −15 | Deacclimation | Root cambium | Alabama |
| 1932 | March | 10 | −7 | Deacclimation | Shoot cambium | Mississippi |
| 2022 | March | 20 | | ——————————— equinox ——————————— | | |
| 1938 | April | 7 | −4 | Deacclimation | Branch cambium | Texas |
| 2007 | April | 8 | −4 | Deacclimation | No injury | Tennessee |
| 2002 | May | 21 | −4 | Deacclimation | Needle burn | North Carolina |
| 1908 | June | 3 | −4 | Deacclimation | No injury | New York |

**Table A2.** Estimated cost of treatments that increase survival at 1000 seedlings ha$^{-1}$.

| Treatment | Cost | Without Treatment | With Treatment | Increase in Survival | Reference |
|-----------|------|-------------------|----------------|---------------------|-----------|
| | \$ ha$^{-1}$ | % | % | % | |
| Correct root gel | 0.03 | 51 | 81 | +30 | [234] |
| Correct root gel—0.33% W/W | 0.15 | 75 | 88 | +13 | [87] |
| Correct clay root dip | 0.25 | 75 | 89 | +14 | [57] |
| No root pruning | 3 | 80 | 76 | +4 | [53] |
| Soaking roots before planting | 10 | 81 | 88 | +7 | [83] |
| Fungicide root dip | 10 | 73 | 85 | +12 | [114] |
| Site preparation burn | 75 | 86 | 94 | +8 | [11] |
| Machine planting—hand plant | 75 | 75 | 86 | +11 | [69] |
| Planted half stem deep—sandhills | 75 | 80 | 90 | +10 | [235] |
| Refrigerator storage—21 days | 126 | 53 | 77 | +24 | [37] |
| Herbicide—herbaceous | 130 | 73 | 92 | +19 | [81] |
| Imidacloprid in planting hole | 130 | 74 | 86 | +12 | [153] |
| Slow careful planting | 160 | 73 | 83 | +10 | [3] |
| Site preparation herbicide | 220 | 89 | 97 | +8 | [236] |
| Carbofuran in planting hole | 230 | 47 | 85 | +38 | [50] |
| Carbofuran treated roots | 230 | 90 | 95 | +5 | [154] |
| Single bedding | 253 | 74 | 90 | +16 | [18] |
| Double bedding | 323 | 75 | 80 | +5 | [237] |
| Subsoiling | 370 | 81 | 88 | +7 | [21] |
| Subsoiling | 370 | 73 | 90 | +17 | [2] |
| 3-in-1 combination plow | 460 | 82 | 96 | +14 | [230] |
| Somatic embryogenic stock | 500 | 77 | 82 | +5 | [238] |
| Mulch | 655 | 37 | 56 | +19 | [137] |
| Sewage sludge | 2000 | 84 | 94 | +10 | [239] |

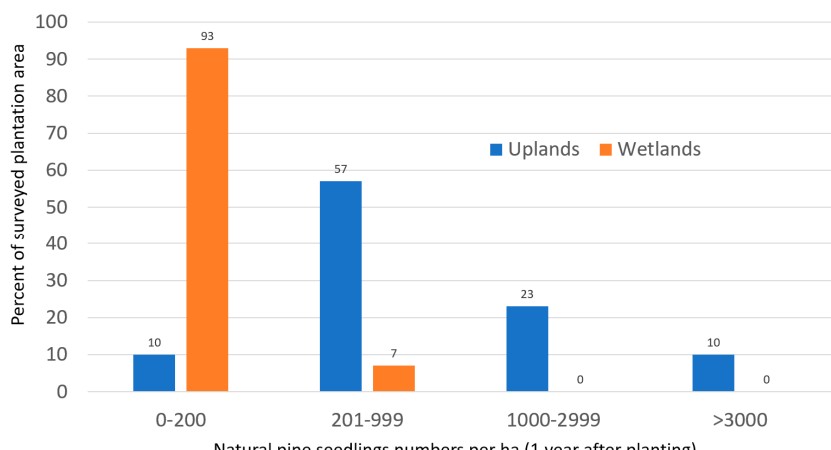

**Figure A1.** A regeneration survey (>4700 ha) indicates 90% of planted upland areas had more than 200 naturally seeded pines ha$^{-1}$. As a result, precommercial thinnings are used so managers can harvest a high percentage of genetically improved stock at the end of the rotation. In 2019, over 24,000 ha of stands were precommercially thinned [8].

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
