# Peer review of "Why Healthy Pine Seedlings Die after They Leave the Nursery"

_forests, doi:10.3390/f14030645_

Round 1
Reviewer 1 Report
The paper is a review of reasons for pine seedling mortality. It is not a a report of a scientific trial but rather a review of reasons for mortality of pine seedlings in plantations. The authors rely on some scientific studies but also many other non-reviewed studies, informal trials, extension bulletins, and anecdotes, some of which are very much out of date. The article would be better suited to publication as a University Extension monograph on reasons for seedling mortality. The article needs extensive editing for length and content and could be much shorter. The recommendations in the article would be more valuable if they were based solely on scientific trials and not on anecdotes or others' opinions.
Specific comments:
Line 20: Keyword should be perhaps “freezing injury”
Figure 1: without any indication of error bars or variance, it is hard to see that Figure 1 represents a general increase.
Line 25: Do we really need the acronym SUS? Maybe just write “southern US”?
Line 29: is there a reference for this statement?
Line 35 and following: It is difficult to follow the argument because many different time periods are used. The overall thesis of the paper seems to be that seedling survival has increased in the past 45 years, but the rainfall data is shown for the past 124 years. I am not clear if the authors are arguing that increased rainfall has led to increased survival or if they are arguing that increased rainfall is not the reason for increased survival.
I don’t understand what Fig. 3 is supposed to show. There does not seem to be any trends within the lines. What is the x axis? Three of the lines more or less overlap in time but the fourth (the green line) occurs later. Is there an argument that survival is increasing with time? If the figure is just used to illustrate the statement that “Even when hand-planting container-grown stock, average mortality can exceed 15% in some years (Figure 3)” having a figure to show this is unnecessary and the paper by McKeand can simply be cited.
Line 61: It seems inappropriate to include papers on Sitka spruce and Douglas fir.
Line 140: The example of Coosa County has nothing to do with the theme of the paragraph on burning.
Figure 4 and others: Error bars would be appropriate on these graphs.
Figure 5. I do not see the utility of this graph. I do not think that the given reason that landowners can easily understand it is valid. I do not see the significance of the regression line. Is the argument that the intercept term is positive proof that burning and bedding is effective? If that is the case, the significance of the intercept term should be tested. It seems that a simple ANOVA of the data would be more revealing. I have the same questions for all the following similar graphs.
Line 176: I assume you mean volume growth of individual trees, which would increase if mortality was high and there was delayed within stand competition.
Figure 6: Same questions as with figure 5. Since the caption implies that the mechanism of mortality involved drought, it would be good to look at survival data vs. rainfall.
Line 205 and following: without knowing the soils in the examples given, the results don’t really mean anything.
Figure 7 is clear and informative but I would like to know if the slopes of the regression lines are significantly different.
Figure 8: The message that exposing roots causes increases in mortality is clear. The inset graph including the lifting date, the rel. humidity, etc. is unnecessary.
Figure 10: The comparison of hand-planted vs. machine-planted seedlings is only valid if seedlings are planted on the same sites. If hand planting is used on more difficult sites, survival would be expected to be lower.
Line 355: If the goal of the paper is to promote the current understanding of how to reduce seedling mortality, why are machine operational statistics from 40 years ago being cited? Has there been no improvement in technology since 1982?
Line 403: This section is on the genetics of the seedlings and should be combined with section 5.4 on bringing genotypes too far north (i.e. off site for the seed source). Are there mapped seed zones for southern pines? If so, these should be referenced.
Line 421: Soaking roots could help if the seedlings are partly dried out. However, in the two examples cited it is not stated whether the seedlings were dried out, nor was there a control of seedlings that were not soaked. No conclusions can be drawn from the examples presented.
Line 483: Does this imply that the planters are burying the foliage? I have never seen this and it does not seem to be good practice.
Line 510: The paragraphs where the authors say that there are two “schools of thought” about whether a practice is helpful or harmful are not helpful to the reader. The authors should carefully consider the sources and produce a recommendation for the reader based on science.
Line 648: The idea that fertilization kills seedlings is very much over-emphasized here. There are thousands of fertilization studies that don’t show any excess seedling mortality. The idea that fertilizer placed in hole 15 cm from the seedling can somehow burn the roots and reduce survival does not make sense.
Line 659: In the cases cited, were there previously issues with root rot at the sites? Without knowing the problem, it is impossible to assess the value of the remedy. Likewise for the next section.
Line 680 Adding a peat wedge to the planning hole seems to be a very uncommon action and probably not worth discussing.
Line 686: Do anti-transpirants reduce growth, since they reduce transpiration and presumably photosynthesis?
Figure 23 Good information but I would like to see the significance of the regression lines (p value). The very low correlation for the AL-M-TN data undercuts the argument that the drought index is a good predictor of mortality.
Figure 24: It would be more interesting to plot % shade vs % survival.
Line 983: The heading is “temperature” but the text describes sunlight.
Line 995: The heading is “high temperature” but the example (15 deg C) is not high by any plant standards.
Figure 28: Secondary Y axis is missing.
Figure 30 shows no relationship between freezing temperatures and seedling survival and can be omitted.
Figure 31 shows no difference between frozen and unfrozen seedings and can be omitted.
Lines 1259 on precommercial thinning is outside the scope of the paper.
Line 1280: The discussion on statistics is outside the scope of this paper.
Author Response
"Please see the attachment."
Line 20: Keyword should be perhaps “freezing injury” Change made
Figure 1: without any indication of error bars or variance, it is hard to see that Figure 1 represents a general increase. As you know, error bars above each are not possible when there is only one value per bar. Survey data is not the same as a research trial with four or more replications. Even with replication, many trials are unable to show a 15% increase in survival as statistically significant (see table 3).
Line 25: Do we really need the acronym SUS? Maybe just write “southern US”? SUS makes for a shorter paper.
Line 29: is there a reference for this statement? Yes. https://rngr.net/publications/tpn/65-2/forest-nursery-seedling-production-in-the-united-states2014fiscal-year-2021/at_download/file
Line 35 and following: It is difficult to follow the argument because many different time periods are used. The overall thesis of the paper seems to be that seedling survival has increased in the past 45 years, but the rainfall data is shown for the past 124 years. I am not clear if the authors are arguing that increased rainfall has led to increased survival or if they are arguing that increased rainfall is not the reason for increased survival.
We say that an increase in rainfall “might help explain some of the increase in rate of survival.” We do not argue that a 5% increase in rainfall caused a 15% increase in survival. Although correlation does not prove causation, this is no reason to hide data from landowners. Our paper is a general review that provides useful information to landowners.
I don’t understand what Fig. 3 is supposed to show.There does not seem to be any trends within the lines. What is the x axis? Three of the lines more or less overlap in time but the fourth (the green line) occurs later. Is there an argument that survival is increasing with time? If the figure is just used to illustrate the statement that “Even when hand-planting container-grown stock, average mortality can exceed 15% in some years (Figure 3)” having a figure to show this is unnecessary and the paper by McKeand can simply be cited.
This figure allows landowner to see the year-to-year variability in survival. Blue line for Texas is lower than Alabama lines and “rainfall for East Texas (1,176 mm) is about 17% less than average rainfall for North Alabama (1,418 mm).” The x-axis helps readers determine the year for each mean survival point.
Line 61: It seems inappropriate to include papers on Sitka spruce and Douglas fir.
We would love to substitute 22 pine points in Figure 24 for the 22 Douglas fir points. If someone had conducted a “rough handling” study on loblolly pine… there would be no need to include the two Sitka spruce references. We cite no papers about rough handling of pine seedlings (since they might not exist). We do cite the good work with spruce and Douglas.
Line 140: The example of Coosa County has nothing to do with the theme of the paragraph on burning.
Coosa County, in central Alabama was in an extreme drought from July 2007 to March 2008 resulting in lower-than-normal survival. “During a period of a severe drought in central Alabama, sites with a burn averaged 73% survival while survival on no burned areas averaged 61% (Figure 4).”
Figure 4 and others: Error bars would be appropriate on these graphs.
We now show LSD values for figures 4, 5. 10 and 31. However, error bars cannot be provided for graphs when cited papers provide only one mean per treatment.
This graph in a 2020 review in Forests did not have standard error bars since the paper was a review and the authors did not have access to replication means. https://www.mdpi.com/1999-4907/11/8/799 This is because the cited reference provided one mean per treatment. Researchers cannot calculate an error bar without knowing each replication mean.
Figure 5. I do not see the utility of this graph. I do not think that the given reason that landowners can easily understand it is valid. I do not see the significance of the regression line. Is the argument that the intercept term is positive proof that burning and bedding is effective? If that is the case, the significance of the intercept term should be tested. It seems that a simple ANOVA of the data would be more revealing. I have the same questions for all the following similar graphs.
Here is the ANOVA for Figure 5. As you can see, the F-value was 88 for treatment.
Line 176: I assume you mean volume growth of individual trees, which would increase if mortality was high and there was delayed within stand competition.
We said “flat disking may increase early volume growth due to a 9% increase in survival, but there may be no increase in volume where disking has no effect on survival [18-20].” In fact, volume growth of individual trees decreases when 13-yr survival increases by 9%. At one loblolly pine site [18], survival increased and volume/ha increased by 11% at age 15 years.
Figure 6: Same questions as with figure 5. Since the caption implies that the mechanism of mortality involved drought, it would be good to look at survival data vs. rainfall.
We see the utility of letting landowners know that ripping soil (before a 160 mm deficit in rainfall) improves survival when average survival is less than 50%. Ripping does little to improve survival at sites where rainfall on non-ripped areas results in survival greater than 90%.
Line 205 and following: without knowing the soils in the examples given, the results don’t really mean anything.
Everyone is entitled to their own opinion regarding the effect of applying over 11,000 kg of lime (before planting) on survival of white pine seedlings. Information on soils (table 2 below) is available from the cited reference. I hope the soil data adds some meaning to the survival results. We find researchers have a tendency to make a Type 2 error when a treatment really does reduce survival by 9 percentage points.
Figure 7 is clear and informative but I would like to know if the slopes of the regression lines are significantly different.
Distance between lines is what is important for this figure…. even if all four slopes were identical!
Figure 8: The message that exposing roots causes increases in mortality is clear. The inset graph including the lifting date, the rel. humidity, etc. is unnecessary.
Data on lift date and relative humidity helps explain to landowners and extension agents why survival was lower for the April 12 planting date. We will not delete data because one reviewer is not interested in the temperature or humidity of the air when seedlings were planted.
Figure 10: The comparison of hand-planted vs. machine-planted seedlings is only valid if seedlings are planted on the same sites. If hand planting is used on more difficult sites, survival would be expected to be lower. We agree. Also, the comparison of hand and machine planted seedlings (at same site) confounds planting depth with planting method. Somewhere in my files, I have unpublished data from Georgia to show better survival with deeper planted stock when both were planted at the same site (but planting depth differed by planting method).
Line 355: If the goal of the paper is to promote the current understanding of how to reduce seedling mortality, why are machine operational statistics from 40 years ago being cited? Has there been no improvement in technology since 1982?
Due to your comment, we added a sentence using 22-yr old survey data from Louisiana. However, “improved” technology that results in increased costs planting does not invalidate the 40-yr old data from Alabama. Although many dollars have been allocated to designing planting machines that are more prone to downtime. In fact, I understand that machine planting in Sweden has declined from about 12% (30 years ago) to <1% now. In contrast, the Whitfield and Reynolds tree planters have proven reliability and machine planting in the SUS is about 40% of the total. These rugged machines are still purchased in the SUS since they can plant many seedlings before needing repair. [We would be happy to cite machine and hand-planting trials that were conducted in the SUS during the 21st century but many young researchers generally stay behind computers and avoid conducting field research. Fortunately, research with machine planting continues in other countries, with some reporting cost per seedling.] When the improvements in technology increase planting costs, smart landowners switch to planting by hand. Automated machines that cost $1,000,000 each are not likely to be used operationally in the SUS. Al Lyons tried out the expensive Plantma machine in Alabama and on the next site they could not get it to work properly.
Line 403: This section is on the genetics of the seedlings and should be combined with section 5.4 on bringing genotypes too far north (i.e. off site for the seed source). Are there mapped seed zones for southern pines? If so, these should be referenced.
The two sections could be combined but we choose to keep them separate. Moving stock north is different than moving genotypes west. Regarding seed zones, citation #77 is sufficient.
Line 421: Soaking roots could help if the seedlings are partly dried out. However, in the two examples cited it is not stated whether the seedlings were dried out, nor was there a control of seedlings that were not soaked. No conclusions can be drawn from the examples presented.
Thanks much. Text now reads… Ideally, roots should not be allowed to dry in storage or during transportation. Since this does occur [78-80], then soaking roots in water might increase the probability of seedling survival (Figure 12) [79,81]. In Pennsylvania, loblolly pine roots were soaked for 24 hours before planting and nine more seedlings survived when compared to nonsoaked seedlings (40 seedlings planted per treatment; 20 per replication). However, soaking roots for 4 hours before planting reduced survival in Mississippi for some unknown reason [82].
Line 483: Does this imply that the planters are burying the foliage? I have never seen this and it does not seem to be good practice.
This graph is about 20 years old. Please send me newer references that contradict our findings.
Machine planters in the SUS bury some pine foliage when planting loblolly or slash pine seedlings. We have published several papers on this topic and know that reducing the rate of transpiration (by top-pruning or planting the root-collar 15 cm deep) can reducing transplanting shock. One reason to publish a general review paper is to make people think about practices they would not recommend to landowners.
Line 510: The paragraphs where the authors say that there are two “schools of thought” about whether a practice is helpful or harmful are not helpful to the reader. The authors should carefully consider the sources and produce a recommendation for the reader based on science.
Landowner should know that one school uses science and the other school relies on traditional recommendations. That way they might learn which school has actually tested a few null hypotheses and which school relies only on logic and tradition.
Line 648: The idea that fertilizer placed in hole 15 cm from the seedling can somehow burn the roots and reduce survival does not make sense.
When fertilizer is applied 15 cm from the seedling, this can increase weed growth. Seedlings planted in sands likely die due to limited soil moisture. We did not say the mortality at this stie was due to burning the roots. In fact, the next sentence said “To reduce the chance of increasing weed growth, some researchers place a low rate of fertilizer below ground at time of planting or several months later.”
Line 659: In the cases cited, were there previously issues with root rot at the sites? Without knowing the problem, it is impossible to assess the value of the remedy. Likewise for the next section.
Pathogens surveyed were listed in cited papers. It is easy for me to assess the value of a 31% increase in seedling using an economic analysis. Please do not make assumptions about the causes of longleaf pine mortality. Not all fungi are root rots. Brown spot is not a root rot. Believing the cause of mortality was due to root rot does not make it easier to determine the value of the improved pine stand. At one time benomyl was registered by EPA for use on roots of pine seedlings and was briefly used operationally on loblolly and longleaf pine. We hope our review will make people stop and think twice about practices they do not recommend.
Line 680 Adding a peat wedge to the planning hole seems to be a very uncommon action and probably not worth discussing.
Placing a peat wedge under bareroot stock is very uncommon. Does everyone already know that the peat cone imbedded with roots is the reason why container stock survives than bareroot stock? Does container stock with 100% sand media have the same survival as container stock with 100% peat? We know that media type affects seedling survival. One reason to write a general review paper is to make people think.
Line 686: Do anti-transpirants reduce growth, since they reduce transpiration and presumably photosynthesis?
Depends on the anti-transpirant, method of application (read the citations) and if the statistical power of the test. Does not reduce growth if the statistical test has low power and cannot detect an 8 cm reduction in height. Even so, survival data are superior to assumptions about photosynthesis and height growth. A height growth increase of 8 cm was significant (using a two-tailed test) for loblolly pine at one site… but at another site, a decrease of 2.6 cm in height was not significant.
Figure 23 Good information but I would like to see the significance of the regression lines (p value). The very low correlation for the AL-M-TN data undercuts the argument that the drought index is a good predictor of mortality.
Landowners are often not interested ANOVA tables or P-values. Researchers that are interested in P-values can find answers in the cited papers. Some researchers may make Type 2 statistical errors when the P-value is 0.11 for a trial with low statistical power. Remember, this general review “is not a report of a scientific trial.” If you really are interested in the P-value…. Read the cited paper where this table is listed. I know that several researchers have ignored the drought index when they planted pine seedlings in Texas and Arizona.
Figure 24: It would be more interesting to plot % shade vs % survival. Thanks… Y axis title changed.
Line 983: The heading is “temperature” but the text describes sunlight. Thanks…. Changes were made.
Line 995: The heading is “high temperature” but the example (15 deg C) is not high by any plant standards. Thanks…. Changes were made.
Figure 28: Secondary Y axis is missing. The secondary Y-axis was labeled “Temperature” and temperature values are listed above the bars.
Figure 30 shows no relationship between freezing temperatures and seedling survival and can be omitted.
We want to show no relationship between freezing temperatures and survival for acclimated stock…. and that mean site survival (x-values) had no effect on the results.
Figure 31 shows no difference between frozen and unfrozen seedings and can be omitted.
We want to show no effect on the no relationship between frozen and unfrozen stock in this trial.
Lines 1259 on precommercial thinning is outside the scope of the paper.
Sometimes mowing wildlings 2 years after planting will also kill some planted pine seedlings.
Line 1280: The discussion on statistics is outside the scope of this paper.
Landowners need to know that researchers often establish survival trials with low statistical power. This is why researchers almost never recommend effective treatments that increase survival by 6%.

Reviewer 2 Report
Thank-you for inviting me to conduct the review of “Why Healthy Pine Seedlings Die After They Leave the Nursery”
Some overall comments:
1. It is long – this is just a comment. I understand reviews tend to be longer, partially due to the increased reference list required. Even though this review has a narrowed focus in terms of factors affecting pine seedling mortality post-nursery in the SUS, it is still a long read.
2. Despite the length it is a fascinating read. Not only is good to see the wealth of data that is available on this topic for a specific region, but more importantly the compilation/consolidation of all this information into one review article. In addition the authors are to be applauded for using available data sets to produce the 1:1 figures for the various factors – this helps visually with any interpretation/understanding.
3. Perhaps what would help the reader (land-owner) would be a summary at the end in the form of a table that lists all the factors discussed that affect survival, together with some ranking/range, and brief manner for mitigation. Something along the lines of Table 3, but expanded. Maybe also some degree of certainty that each factor will reduce mortality.
4. It would also be great to see something within the discussion around the interaction between different factors impacting survival. Research studies may have 2 or more factors, or associated climate data may be linked to time of planting/seedling age or size/soil type etc. Were any of these factors additive, or do they all act independently (unlikely)? Perhaps this is a recommendation for future work?
5. The writing style is between that for a popular article and journal – I have highlighted a few instances where the wording could be improved to be more appropriate for a journal article, but the onus is on the Journal editorial staff to determine if the writing style used for this review is acceptable.
6. Comments have been included on the attached PDF

Author Response
1. Forests has no restrictions on the length of manuscripts, provided that the text is concise and comprehensive
2. Thanks... we also like the 1:1 figures ..
3. Making a table similar to table A4 seems repetitive... and would increase the length. If you can figure out how to calculate a degree of certainty for each factor... please let us know. Certainty of response changes with species and also changes by site and weather.
4. We try not to speculate without data. Weather for a given year does affect survival... but climate is the average of weather for 3 decades or more.
5. "The writing style is between that for a popular article and journal." We prefer a popular review style similar to one recently published in "Forests." Our original submission was a popular review and did not have a materials and method section or a conclusions section. These sections were added at the request of one editor. He wanted our paper to follow a format similar to a systematic review paper (ours is not a systematic review that focused on a single question), I revised the manuscript by including a material and methods section which is routine for general research papers but not for some popular review papers. The current version is not our original style... but it is a style that made it past the journal editors.
Other reviews in "Forests" use a popular writing style... and get published without having to change the style. I assume this means a popular styles are acceptable in the journal known as "Forests." So glad that you made wording suggestions to make it more appropriate for this journal. I am very pleased that Dr. Lamhamedi said our review perfectly meets the objectives of the special issue.
Thanks much for the pdf with great feedback... Your review improved our manuscript.
Regarding figure 33... the paper by Amateis, R.L.; Burkhart, H.E. Rotation-age results from a loblolly pine spacing trial. South. J. Appl. For. also listed spacings out of sequence and we followed their lead. This is why the spacings listed were odd. We changed the graph according to your great suggestion. I also found one typo in the figure as well.
Reviewer 3 Report
The work is very long, literature covers 243 position. Is a metanalysis that covers answear for very important question in Amrican wood indudtry.
Some parts need to be improve:
The discussion and conclusions are out of proportion to the rest of the work.
In the methods section lack information about meta analysis processes. Graphs in the text may have better description to be more imformative. In all text Authors highlight that this results should be easy to reads for landowners but I have daubt of that.
What about the statistic? Not mention in M and M section but were mention in the description of some figures not all that should be given this information.
Authors could consider sumarize their findings into grapgs for popularisation this result into groupe of landowners.
Fig 1. What sort of information cover this figure. When the germination were done in the year of harvest or now. Whay there are reange of yerars and at the end one specific year.
Fig 4 Authors not explained what mean numbers on the bar and above them.
Table 1 Second row should be deleyed and the information shoud be transfer to first row.
Author Response
- Forests has no restrictions on the length of manuscripts, provided that the text is concise and comprehensive
Some parts were improved.
Regarding statistics, the cited references used different methods. Therefore, the methods used were not the same for all figures and the few interested in statistical methods employed can download many of the citations from the web.
A simple regression was developed for several figures? Do I need to include a textbook reference on how to calculate a simple regression equation?
[Note... the following is not clear ... "but were mention in the description of some figures not all that should be given this information."]
Not sure how to respond to the following. "Authors could consider sumarize their findings into grapgs for popularisation this result into groupe of landowners."
Figure 1 regards seedling survival after transplanting.... not seed germination.
Figure 4 Numbers above bars are same as Y-values
Table 1 ,,,Second row was deleted
Reviewer 4 Report
I express my compliments to the authors for this manuscript.
Novelty: the question is original and well-defined and the results provide an advancement of the current knowledge.
Scope: the work fit the journal scope.
Significance: the results are interpreted appropriately and are significant. All conclusions are justified and supported by the results. Quality: the article is written in an appropriate way and the data and analyses are presented appropriately.
Scientific Soundness: the study is correctly designed and technically sound. The analyses are performed with the highest technical standards. The data is robust enough to draw conclusions and the methods, tools, software, and reagents arre described with sufficient details to allow another researcher to reproduce the results.
Interest to the Readers: the conclusions are interesting for the readership of the journal.
English Level: the English language is appropriate and understandable.
Author Response
Novelty: the question is original and well-defined and the results provide an advancement of the current knowledge. [THANKS}
Significance: the results are interpreted appropriately and are significant. All conclusions are justified and supported by the results. Quality: the article is written in an appropriate way and the data and analyses are presented appropriately.
[Thanks.... glad you could follow the statistics]
{Thanks for your feedback}
Round 2
Reviewer 1 Report
I read the author's comments on my review. I do not think the lead author addressed my comments to any significant extent. His responses included references which were not in the manuscript. His arguments rely on many unreplicated studies and anecdotes. I do not think the paper is suitable for publication in its current revised form.
Author Response
Thanks for taking the time to review our paper. You believe our review would be better suited as a University Extension monograph.